# Local weakening of cell-extracellular matrix adhesion triggers basal epithelial tissue folding

Andrea Valencia-Expósito [1✉], Nargess Khalilgharibi[2,3], Ana Martínez-Abarca Millán [1], Yanlan Mao[2,3] & María D Martín-Bermudo [1✉]

## Abstract

During development, epithelial sheets sculpt organs by folding, either apically or basally, into complex 3D structures. Given the presence of actomyosin networks and cell adhesion sites on both sides of cells, a common machinery mediating apical and basal epithelial tissue folding has been proposed. However, unlike for apical folding, little is known about the mechanisms that regulate epithelial folding towards the basal side. Here, using the *Drosophila* wing imaginal disc and combining genetic perturbations and computational modeling, we demonstrate opposing roles for cell-cell and cell-extracellular matrix (ECM) adhesion systems during epithelial folding. While cadherin-mediated adhesion, linked to actomyosin network, regulates apical folding, a localized reduction on integrin-dependent adhesion, followed by changes in cell shape and reorganization of the basal actomyosin cytoskeleton and E-Cadherin (E-Cad) levels, is necessary and sufficient to trigger basal folding. These results suggest that modulation of the cell mechanical landscape through the crosstalk between integrins and cadherins is essential for correct epithelial folding.

**Keywords** Integrins; Actomyosin; Cadherins; Constricting Forces
**Subject Categories** Cell Adhesion, Polarity & Cytoskeleton; Development

## Introduction

Tissue folding is a fundamental process that sculpts simple flat epithelia into complex 3D organ structures (Heisenberg and Bellaiche, 2013). During most tissue folding events, an increase in actomyosin activity in a specific group of cells followed by a change in cell shape serves as an initiating point for folding (reviewed in Leptin et al, 1989). Defects in local activation of actomyosin in the appropriate cells during development results in imprecise tissue folding and defective organogenesis (Martin et al, 2010). Hence, deciphering the mechanisms that regulate precision of tissue folding in space and time is crucial to further comprehend morphogenesis.

This knowledge could also be applied to engineer folded systems, significantly advancing the field of tissue engineering.

During morphogenesis, tissue folding can initiate either on the apical or basal surface. Apical tissue folding has long been studied and commonly attributed to apical constriction, mediated by forces generated by the contraction of an actomyosin network and the transmission of these forces to cell-cell adhesion sites (Christodoulou and Skourides, 2015; Martin and Goldstein, 2014; Martin et al, 2009). Additional mechanisms include basolateral contractility (Sherrard et al, 2010), differential growth rates (Tozluoglu et al, 2019), microtubule network remodeling (Takeda et al, 2018) and basal and lateral tension (Sui et al, 2018). However, even though basal folding is widespread and required for diverse morphogenetic events in both vertebrate and invertebrate systems (Dong et al, 2011; Gutzman et al, 2008; Holz et al, 2017; Martinez-Morales et al, 2009), little is known about the mechanisms that mediate this crucial developmental process. Given the presence of actomyosin networks and sites of cell-extracellular matrix (ECM) adhesion on the basal side of epithelial cells, the question remains as to whether, analogous to what happens during apical constriction, contraction of basal actin-myosin networks and their attachment to cell-ECM adhesion sites may constitute a force-generating mechanism underlying basal constriction. A recent study has identified a cell-cell adhesion complex containing E-Cadherin (E-Cad) at the basal most-region of the lateral membrane in different *Drosophila* epithelial tissues, including larval wing imaginal discs, which can be modulated by cell-ECM interactions (Kroeger et al, 2024). This has been proposed to regulate morphogenetic processes, although this needs to be yet demonstrated.

The larval *Drosophila* wing imaginal disc provides an excellent in vivo model to investigate the mechanisms controlling basal epithelial folding during development. The wing disc is a sac-like structure composed of two opposing cell layers, a peripodial membrane (PM) and the wing disc proper, henceforth, wing disc (Appendix Fig. S1). The wing disc is a pseudostratified columnar epithelium that attaches on its basal side to a basement membrane (BM). The wing disc can be divided into four morphological distinct regions that will differentiate into consistent adult structures: the wing pouch will give rise to the wing blade; the proximal wing and wing hinge will form structures at the base of the wing; the notum will give rise to the back of the fly in the thorax and the PM will form the pleura. Morphogenesis of the wing disc

[1]Centro Andaluz de Biología del Desarrollo CSIC-Univ. Pablo de Olavide, Sevilla 41013, Spain. [2]Laboratory for Molecular Cell Biology, University College London, Gower Street, London WC1E 6BT, UK. [3]Institute for the Physics of Living Systems, University College London, Gower Street, London WC1E 6BT, UK. ✉E-mail: andrea.ve06@gmail.com; mdmarber@upo.es

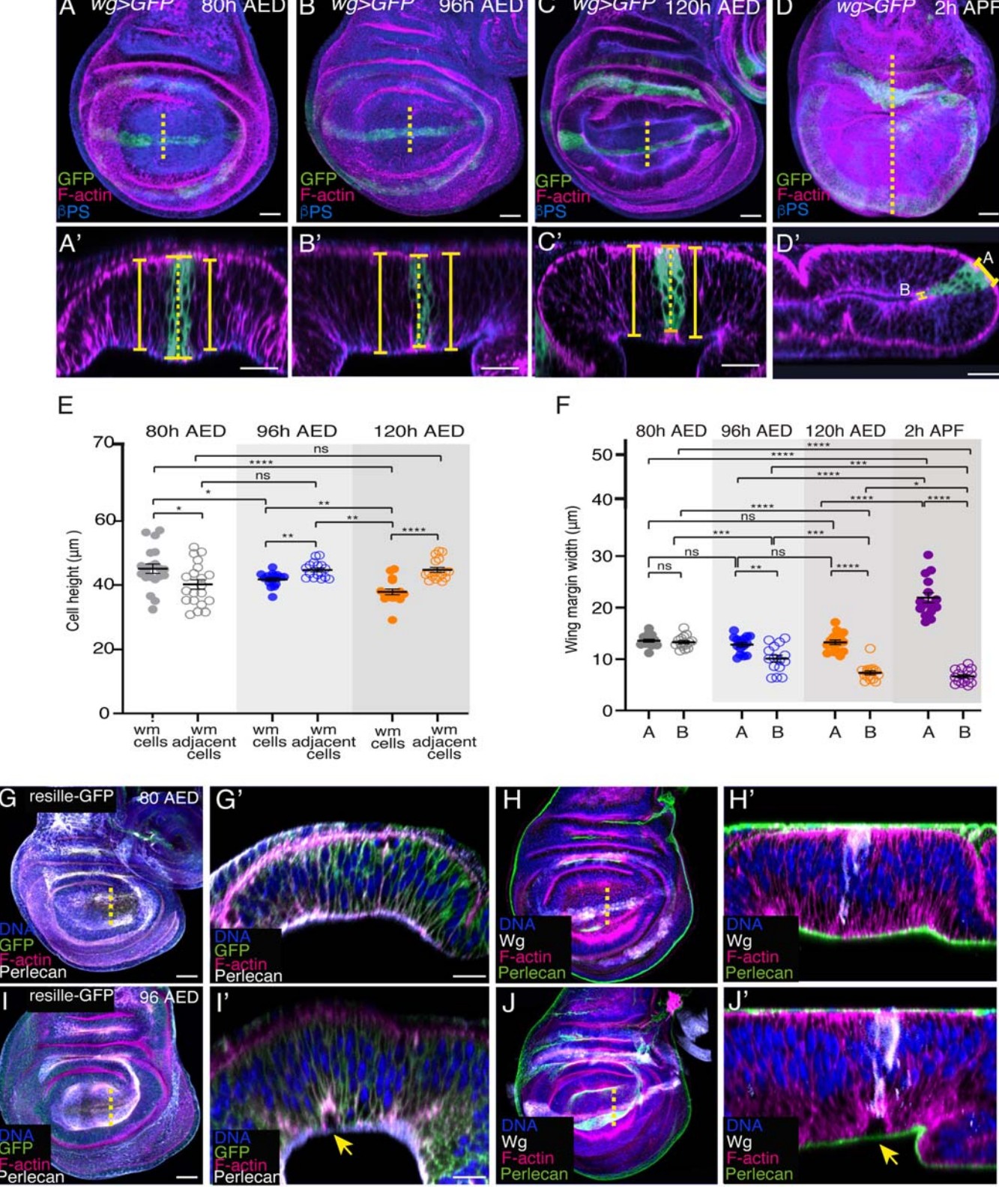

◄ **Figure 1. Wing margin cells shorten and detach from the BM during development.**

(A–D') Confocal views of wing imaginal discs throughout third-instar larvae (A–C') and at 2 h APF (D, D'), stained with anti-GFP (green), anti-βPS (blue) and the F-actin marker Rhodamine Phalloidin (magenta). (A'–D') Confocal YZ cross-sections along the yellow dotted lines shown in (A–D). Brackets indicate cell height in the wing margin (dotted line) and in ventral and dorsal domains (straight line). (E) Quantification of cell height of wing margin and adjacent cells, at different larval developmental stages. Multiple Mann–Whitney U test from left to right: *$p = 0.0375$, *$p = 0.0432$, ****$p = 1.0e{-}4$, **$p = 0.003$, **$p = 0.0014$, **$p = 0.0032$, ****$p = 1.8e{-}5$, ns not significant. Error bars represent the mean ± SEM. (F) Quantification of apicolateral (AL) and basolateral (BL) wing margin width at different developmental stages. Multiple Mann–Whitney U test from left to right: ***$p = 0.00053$, ****$p = 6.2e{-}12$, **$p = 0.00309$, ****$p = 3.5e{-}13$, ****$p = 5.1e{-}13$, ***$p = 0.00235$, ****$p = 1.5e{-}7$, ****$p = 1.9e{-}10$, ****$p = 2.7e{-}7$, ***$p = 0.003$, *$p = 0.02414$, ****$p = 1.0e{-}10$, ns not significant. Error bars represent the mean ± SEM. (G–J) Confocal views of wing discs expressing the membrane marker resille-GFP at 80 h AED (G, H) and 96 h AED (I, J) stained with anti-GFP (green in G, G', I, I'), the F-actin marker Rhodamine Phalloidin (magenta), the nuclear marker Hoechst (DNA, blue), anti-perlecan (white in G, G', I, I' and green in H, H', J, J'). (G', I', H' and J') Confocal YZ cross-sections along the white dotted lines shown in (G), (I), (H) and (J), respectively. Yellow arrows in (I') and (J') point to cell detachment from the BM in the wing margin region. Scale bar in all panels, 30 μm. At least 16 wing discs were assessed over three independent experiments. Source data are available online for this figure.

involves the formation of several folds, both apical and basal (Cohen, 1993). The prospective hinge region forms three stereo-typic apical folds. In contrast, the wing pouch folds basally along the wing margin, a stripe of cells in the middle of the pouch, which promotes the transformation of the single-layered columnar epithelium to a flattened bilayer during disc eversion (Fristrom and Fristrom, 1993). This process of folding occurs at the onset of metamorphosis, around 2 h after puparium formation (2 h AP). At this stage, wing margin cells reduce their height and expand and decrease their apical and basal surfaces, respectively, thus adopting a wedge shape (Fristrom and Fristrom, 1993). Folding in the hinge region has been studied and proposed to be driven by a local decrease in basal tension and an increase in lateral tension, but not by apical constriction (Sui et al, 2018). However, little is known about the mechanical mechanisms that drive basal folding along the wing margin.

In this work, taking a multidisciplinary approach, combining genetic perturbations and computational modeling, we show that wing disc folding along the wing margin is initiated by a precise sequence of events in wing margin cells at the end of the larval period, starting with a reduction in integrin levels, followed by an increase in basolateral contractility and cell shortening. Thus, our results demonstrate opposing roles for cell-cell and cell-ECM adhesion during epithelial folding. While cadherin-mediated adhesion, linked to actomyosin network, regulates apical folding, a reduction in integrin-dependent adhesion, coupled to an increase in basolateral actomyosin levels, is required to initiate and carry out correct basal folding. Furthermore, we show that *Drosophila* E-cad (DE-Cad) is enriched basally in wing margin cells and that the ectopic reduction of integrin levels is sufficient to trigger a basal increase in DE-Cad levels. Based on these results, we propose a model for basal folding in which the basal side of the cells would utilize the same players used for apical folding, an active actomyosin network and DE-Cad adhesion sites, to induce the necessary cell shape changes. This requires and is triggered by a downregulation of cell-ECM interactions mediated by integrins.

## Results

### Characterization of the process of basal folding of the *Drosophila* wing disc along the wing margin

To better understand basal folding, we examined the changes in cell shape that accompanied the folding of the wing disc along the wing margin. This has been proposed to occur at the onset of metamorphosis, around 2 h after puparium formation (2 h AP), when wing margin cells undergo cell shape changes (Fristrom and Fristrom, 1993). However, here we found that some of these changes could already be appreciated during larval stages, and more precisely, during the last of the three larval stages, the third-instar larvae (L3) (Fig. 1).

To analyze possible changes in cell height, we measured the height of wing margin and adjacent cells, throughout L3 development. To clearly distinguish wing margin cells from the rest of the wing pouch cells, we made use of a Gal4 line inserted in the wing margin cell fate determinant gene *wingless* (*wg*Gal4) to direct the expression of the membrane marker CD8-GFP (*wg* > GFP) in wing margin cells (Couso et al, 1994) (Fig. 1). To measure cell height at the wing margin and adjacent cells, we visualized cell boundaries using Rhodamine-Phalloidin (Rh-Ph) that labels F-actin and therefore the cell cortex. We found that while at early L3, 80 h after egg deposition (80 h AED), wing margin cells were slightly taller than dorsal and ventral adjacent cells (Fig. 1A,A',E), from this stage onwards, the height of wing margin cells decreased progressively (Fig. 1B,B',C,C',E), so that by late L3 stage (120 h AED, Fig. 1C,C',E), wing margin cells were significantly shorter than adjacent cells, resulting in a local indentation of the tissue at its basal side (Fig. 1C').

Folding of the wing disc along wing margin cells during pupal stages has been shown to involve, not only a decrease in their height, but also an expansion and a reduction of their apical and basal surfaces, respectively (Fristrom and Fristrom, 1993). As we have shown here that the reduction in the height of wing margin cells already began at L3, we next decided to check whether their apical and basal surfaces also experienced any changes at this larval stage. The pseudostratified nature of the wing disc epithelium makes it difficult to follow the width of the apical and basal sides of a single cell. Therefore, we measured the width of the wing margin region, marked by *wg* > GFP. We found that at 80 h, the width of the wing region at its apical side (A) was similar to that of the basal side (B) (Fig. 1A,A',F). In addition, we found that, while the apical width did not change during L3, the basal width decreased progressively (Fig. 1A–C',F). Finally, we found that at white pupal stage (2 h APF), when the wing disc has folded, the apical and basal width of the wing margin region had increased and decreased, respectively (Fig. 1D,D',F), consistent with the acquisition of a wedge shape at this stage (Fristrom and Fristrom, 1993).

Altogether, our results show that wing margin cells undergo cell shape changes already at larval stages. Next, we tested whether

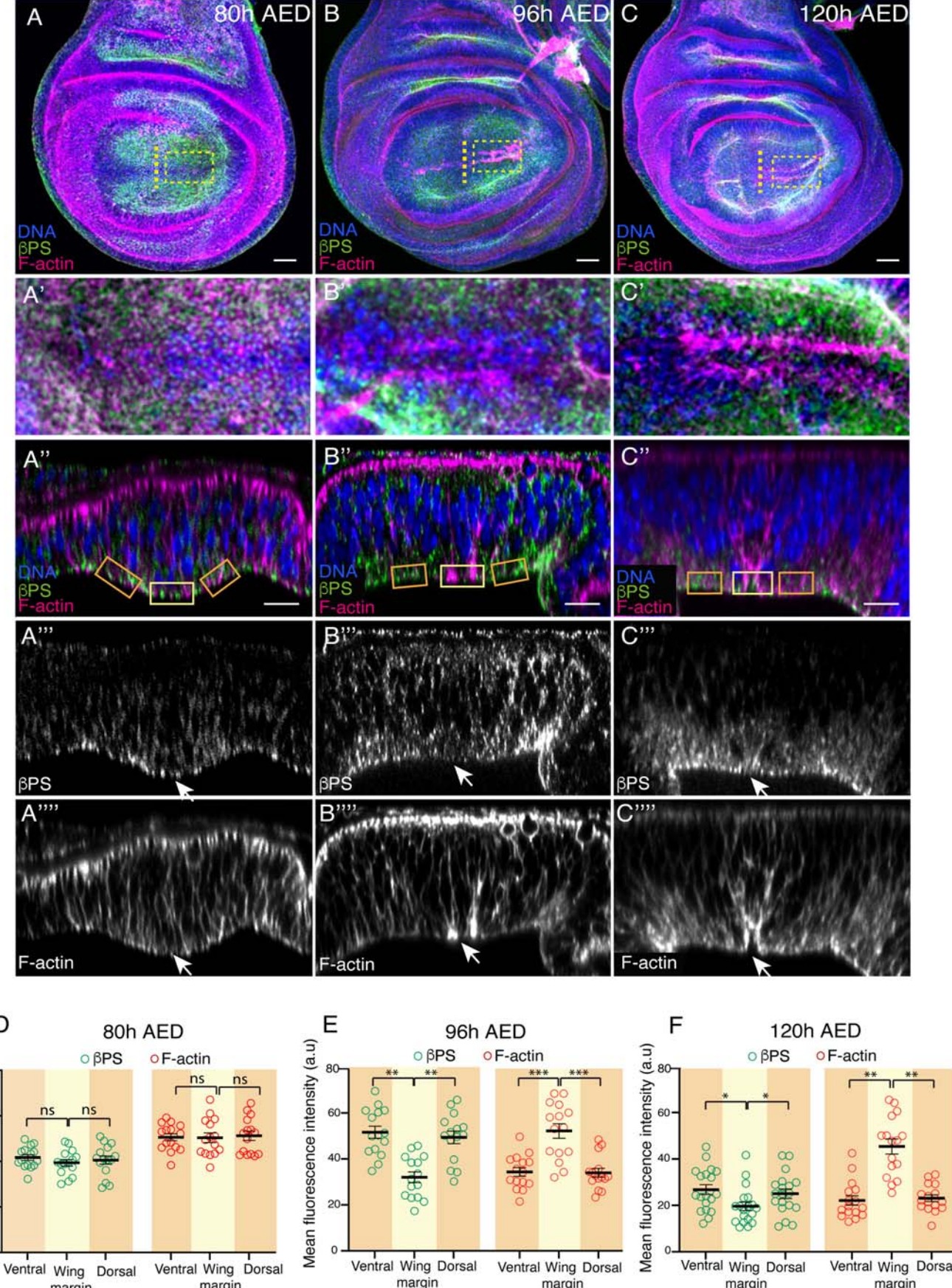

**Figure 2.  β-integrin and F-actin distribution in the wing margin changes over development.**

(**A–C''''**) Confocal views of wing imaginal discs from early to late third-instar larvae stained with anti-βPS (green in **A–C''**, white in **A'''–C'''**), the F-actin marker Rhodamine Phalloidin (magenta in **A–C''**, white in **A''''–C''''**) and the nuclear marker Hoechst (DNA, blue in **A–C''**). (**A–C**) Maximal projections of control wing disc of 80 h AED (**A**), 96 h AED (**B**) and 120 h AED (**C**). (**A'–C'**) Basal surface views of the regions specified in the yellow boxes in (**A–C**). (**A''–C''''**) Confocal YZ cross-sections along the yellow dotted lines shown in (**A–C**). White arrows in (**A'''–C''''**) point to the wing margin region. (**D–F**) Quantification of βPS and F-actin levels in control wing discs of the designated developmental time points in the regions framed in (**A''**), (**B''**) and (**C''**), yellow and orange denote wing margin region and adjacent cells, respectively. Multiple Mann–Whitney U test from left to right: (**D**) ns not significant, (**E**) **$p = 0.0083$, **$p = 0.0014$, ***$p = 1.6e{-}4$, ***$p = 1.5e{-}4$, (**F**) *$p = 0.037$, *$p = 0.0401$, **$p = 0.0025$, **$p = 0.0036$. Error bars represent the mean ± SEM. Scale bar in all panels, 30 μm. At least 15 wing discs were assessed over three independent experiments. Source data are available online for this figure.

these changes were accompanied by a detachment of the cells from the BM. To do this, we used the cell membrane marker Resille-GFP (Morin et al, 2001) and an antibody against Perlecan, a component of the BM of wing imaginal discs (Pastor-Pareja and Xu, 2011). We found that indeed late L3 wing margin cells detached from the BM (Fig.1G–J').

## Dynamics of β-integrin, F-actin, and non-muscle Myosin II distribution in the wing margin during basal fold initiation

The *Drosophila* genome contains two β-integrin subunits, βPS and βν (reviewed in Brown, 1993; Yee and Hynes, 1993). As the βPS subunit, encoded by the gene *myospheroid* (*mys*), is the only β chain present in the wing disc, we analyzed integrin distribution using an antibody against this subunit. We found that at early L3, integrin concentrated basally in all columnar cells (Fig. 2A–A''',D). By mid L3, integrin levels dropped specifically at the basal side of cells in the wing margin region (Fig. 2B–B''',E), a reduction that was maintained at late L3 (Fig. 2C–C''',F). The distribution of F-actin in wing margin cells was also dynamic. At early L3 stage, F-actin was consistently associated to the cell cortex in all columnar pouch cells, with higher intensity along the apical surface (Fig. 2A,A' A'''',D). At mid L3 stage, F-actin accumulated basolaterally specifically in wing margin cells (Fig. 2B,B',B'''',E). This accumulation was maintained at late L3 stage (Fig. 2C,C',C'''',F). Next, we analyzed the distribution of activated non-muscle MyosinII, hereafter MyoII. In order to do this, we used an antibody that specifically recognizes the *Drosophila* homolog of the MyoII regulatory light chain, *spaghetti squash* (*sqh*), when it is phosphorylated at the activating Ser-21 (pSqh). We found that pSqh dynamics were similar to that of F-actin, changing from a homogeneous distribution at early stages to a basolateral accumulation at mid L3 stages (Appendix Fig. S2A,B''').

To analyze in more detail the distribution of integrins and F-actin on the basal side of cells in the wing margin region, we performed super-resolution microscopy of early and mid L3 wing discs. This time, integrin and MyoII expression were visualized using transgenic flies carrying either a green fluorescent protein (GFP) inserted into the *mys* locus or a GFP fused to *sqh* (Martin et al, 2009). We found that at early L3, integrin expression, F-actin and Sqh organized uniformly throughout the wing pouch (Fig. EV1A–A''',C–C'''). However, at mid L3, the levels of integrin, F-actin and Sqh in wing margin cells differed from those of their neighbors. Thus, while integrin levels decreased (Fig. EV1B'–B''',E), basolateral F-actin and Sqh levels increased (Fig. EV1B'–B''',D'–D''',E). In addition, F-actin seemed to

reorganize and accumulated in bright spots specifically in wing margin cells (Fig. EV1B''').

The nuclei of columnar pouch cells at mid L3 exhibit an apically-biased position (Nematbakhsh et al, 2020). This nucleus asymmetrical spatial distribution has been shown to depend on actomyosin contractility and contribute to wing disc shape at a tissue level (Nematbakhsh et al, 2020). As wing margin cells show increased actomyosin levels compared to their neighbors, we hypothesized that their nuclei would be more predisposed to adopt a more apical position than adjacent cells and found this to be the case (Fig. EV2A–A''',D).

These results show that basal fold initiation is accompanied by a reduction in integrin levels and basolateral actomyosin accumulation.

## Modeling the implications of changes in integrin levels and basolateral actomyosin accumulation on basal fold initiation

To ask whether the local reduction in integrin levels and basolateral actomyosin accumulation could initiate basal folding, we adapted a previously developed Finite Element Model (Tozluoglu et al, 2019) and simulated a cross-section of the columnar epithelium perpendicular to the DV boundary (Fig. 3A). The epithelium consisted of an apical actin layer and cell body layers. To account for the integrin adhesions, we included a thin layer on the basal side of the columnar epithelium, representing a 'gap layer of integrins' between the cells and BM (Fig. 3B). To simulate changes in the wing margin, the properties of elements in the middle of the tissue accounting for ~10% of its width were modified. Since the elements in our model are elastic, the integrin adhesion layer (symbolized in a greenish-brown color in the figures) could be considered as a layer of springs whose stiffness represent adhesion strength. In this context, we simulated a decrease in integrin levels as a reduction of the stiffness of the integrin adhesion layer. Phosphorylation of myosin II is required to generate contraction and is therefore often used as a readout of contractility (Tan et al, 1992). Therefore, in our simulations, we implemented basolateral accumulation of actin and phosphorylated MyoII by reducing the preferred height of the most basal elements of the cell body layer in the wing margin region, which would create contractility in the basolateral region. When we reduced the adhesion strength and applied basolateral contractility simultaneously in the wing margin region in our simulations, we found that the cell layer deformed on its basal side, suggesting that the two processes together are sufficient to induce basal folding in the wing margin (Fig. 3C, Movie EV1).

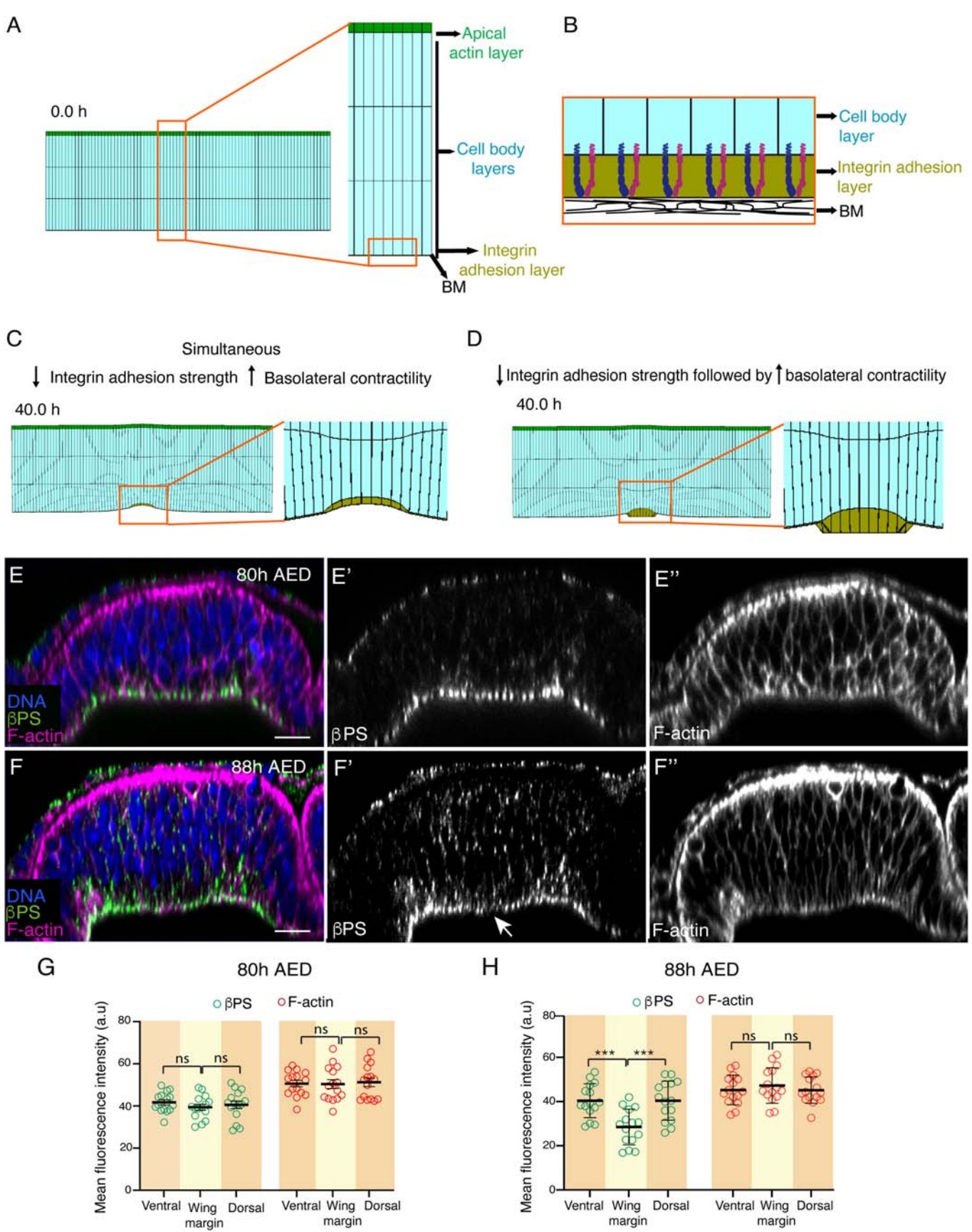

**Figure 3.   Integrin downregulation precedes F-actin changes in wing margin cells.**

(A) Initial simulation at 0 h, showing a cross-section of the wing disc columnar epithelium perpendicular to the DV boundary. On the right, a close-up of the cross-section showing the apical actin layer (green), three cell body layers (cyan) and one integrin adhesion layer (greenish-brown). (B) Interpretive scheme of the integrin adhesion layer framed in orange in (A). Layers' thickness in the scheme are not to scale. (C, D) Snapshot of simulation when basolateral contractility and reduction of integrin adhesion strength were applied simultaneously (C) or when the strength of the integrin adhesion was decreased prior to application of basolateral contractility (D). Magnifications of the region framed in the snapshots are also shown. (E–F") Confocal YZ cross-sections of control wing disc at 80 h AED (E) and 88 h AED (F) stained with anti-βPS (green in **E, F** and white in **E', F'**), Rhodamine Phalloidin to detect F-actin (magenta in **E, F** and white in **E", F"**) and the nuclear marker Hoechst (DNA, blue in **E, F**). (G, H) Quantification of βPS and F-actin levels in control wing discs of the designated developmental time points in the regions framed in **E** and **F** (orange and yellow boxes). Multiple Mann–Whitney U test from left to right: (G) ns not significant, (H) ***$p = 8.8e{-}4$, ***$p = 9.3e{-}4$, ns not significant. Error bars represent the mean ± SEM. Scale bar in all panels, 30 μm. At least 15 wing discs were assessed over three independent experiments. Source data are available online for this figure.

Our experiments have shown that wing margin cells detach from the BM (Fig. 1). In our model, as the layers cannot detach from each other, the detachment of cells from the BM is represented as a thickening of the integrin adhesion layer, where the upper side of this layer in contact with the cells is deformed, while the other side in contact with the BM remains undeformed and flat. However, we found in our simulations that even though a thickening of the integrin adhesion layer was produced, both sides of the layer deformed (Fig. 3C). We therefore asked whether the changes in integrin and basolateral actomyosin levels followed a temporal order, rather than occurring simultaneously. To test this, we used the model to explore a temporal sequence and found that when we decreased the strength of the integrin adhesion layer before applying the basolateral contractility, only the side of the integrin adhesion layer contacting the cells deformed (Fig. 3D, Movie EV2), as seen in vivo. This suggested that in the wing disc, the integrin adhesions should weaken before basolateral actomyosin accumulation occurs to allow for proper basal deformation and detachment from the BM. To test this model prediction, we analyzed integrins expression and F-actin distribution in wing margin cells of an intermediate stage between early and mid L3, at 88 h AED. We found that while levels of F-actin were similar to those found at early stage (Fig. 3E,E",F,F",G,H), integrin levels had already reduced (Fig. 3E,E',F,F',G,H). This result strengthens the idea that a reduction in cell-BM adhesion is the first step triggering the changes in actomyosin organization and cell shape that initiate basal folding.

Altogether, these results suggest that, in contrast to what happens during apical folding where cell-cell adhesion is required to initiate and complete the process, during basal folding, a reduction in cell-BM adhesion is required as a first step to trigger the process.

## Downregulation of integrin levels is sufficient to trigger the changes in F-actin levels and cell shape that accompany initiation of basal wing disc folding

To test that indeed a local reduction in integrin levels could initiate basal folding, we analyzed experimentally the consequences of reducing integrin levels in an ectopic location in the wing disc. For this, we used the *patched*-Gal4 line (*ptcG4*), which drives high levels of expression in a stripe of cells along the anterior/posterior (A/P) boundary of the wing pouch (Ingham et al, 1991). We have recently found that reducing integrin expression in the wing disc epithelium induces caspase-dependent cell death, via induction of expression of the proapoptotic gene *hid* (Valencia-Exposito et al, 2022). Furthermore, expression of an RNAi against *hid* suppressed the cell

death phenotype due to loss of integrin function (Valencia-Exposito et al, 2022). Thus, we co-expressed in the *ptc* domain RNAis against *mys* and *hid* (*ptc>mys^RNAi;hid^RNAi*) and found that this led to a basolateral accumulation of F-actin (Fig. 4A,B",C,D) and p-Sqh (Fig. EV2A–F), cell shortening, the formation of an indentation, basal F-actin reorganization (Fig. 4A'–A"',B'–B",E) and an apical bias position of the nuclei (Fig. EV2B,B"',D), all of which resembled that happening in control wing margin cells.

As mentioned in the introduction, actomyosin generated forces need to be transmitted to the cell membrane to cause cell shape changes, thereby tissue bending (Christodoulou and Skourides, 2015; Martin and Goldstein, 2014; Martin et al, 2009). However, if cell-ECM junctions are downregulated and cell-cell junctions are not present on the basal side of the wing margin cells, how are the forces generated by the actomyosin cytoskeleton transmitted to the plasma membrane? A basal shift of adherens junctions in a group of cells has already been proposed to facilitate epithelial folding before the initiation of dorsal transverse folds during *Drosophila* gastrulation (Wang et al, 2012). We have previously shown that removing integrin function from wing margin cells does not affect their apical-basal polarity, as seen with an antibody against the apical marker aPKC (Martinez-Abarca Millan and Martin-Bermudo, 2023). Next, we analyzed the distribution of *Drosophila* E-cadherins (DE-Cad) in wing disc cells and found an enrichment of basal DE-Cad puncta specifically in wing margin cells (Fig. 5A–A"',B). In addition, we found that the local ectopic reduction of integrin levels created in *ptc>mys^RNAi; hid^RNAi* wing discs was sufficient to induce an increase in basal DE-Cad levels (compare Fig. 5C–C"' and D with E–E"' and F). Furthermore, downregulation of DE-Cad levels specifically in wing margin cells preventing both the basolateral accumulation of F-actin and the formation of a local basal indentation (Fig. 5G–G"',H,I).

During *Drosophila* gastrulation, mesodermal cells undergo apical constriction to form the ventral furrow, through activation of MyoII, by an apically localized Rho-associated kinase (Rok) (Dawes-Hoang et al, 2005). Activated MyoII forms an actomyosin complex at the medial apical cortex that pulls adherens junctions inwards, causing a reduction in apical area (Martin et al, 2009). In addition, a relation between integrins and the Rho-Rok signaling pathways has been reported in *Drosophila* follicle cells. In this case, integrins positively control the intensity and oscillation of medio-basal ROK and MyoII signals (Qin et al, 2017). As we show here that, during basal folding, integrins negatively regulate F-actin, activated MyoII and DE-cad in wing margin cells, we next decided to test Rok distribution in these cells. In order to examine Rok localization, we used an endogenously N-terminally tagged Rok with the bright mNeon-Green (mNG) fluorescent protein

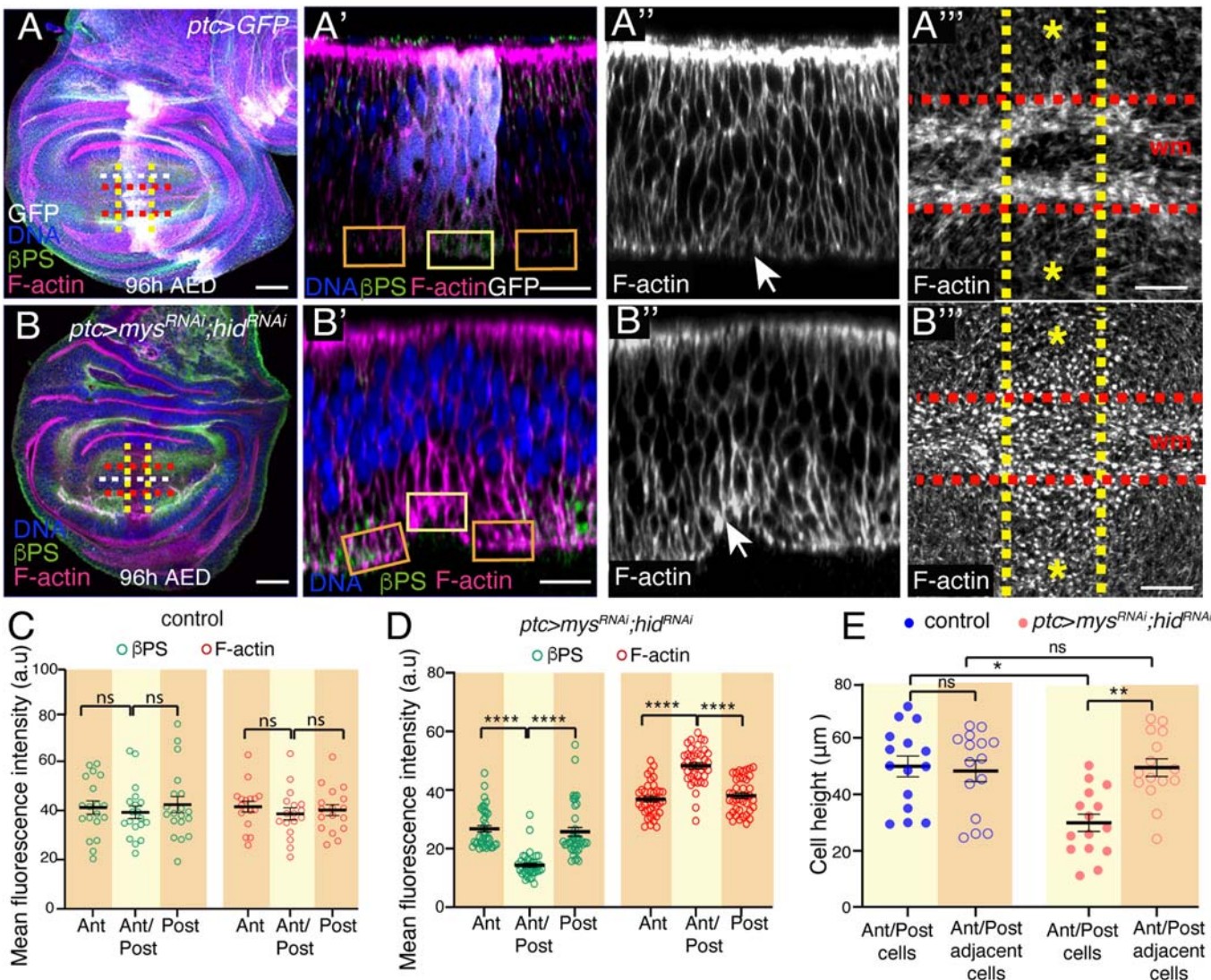

**Figure 4. Ectopic reduction of integrin levels induces actin reorganization and cell shortening.**

(A–B") Confocal views of third-instar wing imaginal discs stained with anti-GFP (white), anti-βPS (green), Rhodamine Phalloidin to detect F-actin (magenta in **A**, **A'**, **B**, **B'**, white in **A"**, **A"'**, **B"**, **B"'**) and the nuclear marker Hoechst (DNA, blue in **A**, **A'**, **B**, **B'**). (**A**) Control wing disc. (**B**) Wing disc co-expressing RNAis against *mys* and *hid* under the control of the *ptcGal4* (*ptc>mys^RNAi;hid^RNAi*). (**A'**, **A"**, **B'**, **B"**) Confocal XZ cross-sections taken along the white dotted lines shown in (**A**, **B**). (**A"'**, **B"'**) Super-resolution images of XY sections taken in the region between the yellow and red dotted lines in (**A**, **B**). (**C**, **D**) Quantification of βPS and F-actin levels in the regions framed in (**A'**) and (**B'**) (orange and yellow boxes) in control (**C**) and *ptc>mys^RNAi;hid^RNAi* (**D**) wing discs. Multiple Mann–Whitney U test from left to right: (**C**) ns not significant, (**D**) ****$p = 5.2e{-}14$, ****$p = 5.8e{-}9$, ****$p = 4.9e{-}12$, ****$p = 4.2e{-}10$. Error bars represent the mean ± SEM. (**E**) Quantification of the height of wing margin and adjacent cells in control and *ptc>mys^RNAi;hid^RNAi* wing discs. Multiple Mann–Whitney U test from left to right: *$p = 0.0183$, **$p = 0.007$, ns not significant. Error bars represent the mean ± SEM. At least 15 wing discs were assessed over three independent experiments. Scale bar in all panels, 30 μm. Source data are available online for this figure.

(mNG-Rok), which was found to be enriched in the apical region of different epithelial cells, including larval eye disc cells (Sidor et al, 2020). Similarly, here we detected an apical enrichment of mNG-Rok in all wing disc cells (Fig. EV3A–A"'). However, interestingly, we also found a specific enrichment of mNG-Rok in the basal side of wing margin cells (Fig. EV3A,A',A"',D). Furthermore, ectopic removal of integrin was sufficient to induce this basal accumulation (Fig. EV3B,C"',E,F).

Altogether, our results show that a reduction in integrin levels is sufficient to induce basal accumulation of RoK, F-actin, pSqh and DE-Cad. In addition, they suggest that mechanisms driving basal

folding may be similar to those underlying apical folding and be governed by a basal activation of an actomyosin network, which, by pulling on adherens junctions, causes a reduction in the basal area. Interestingly, we show this requires first a downregulation of cell-ECM interactions.

## Basolateral actomyosin accumulation is required for basal fold initiation in the wing margin cells

We next analyzed the contribution of basolateral actomyosin to the initiation of basal epithelial folding. To do this, we first challenged

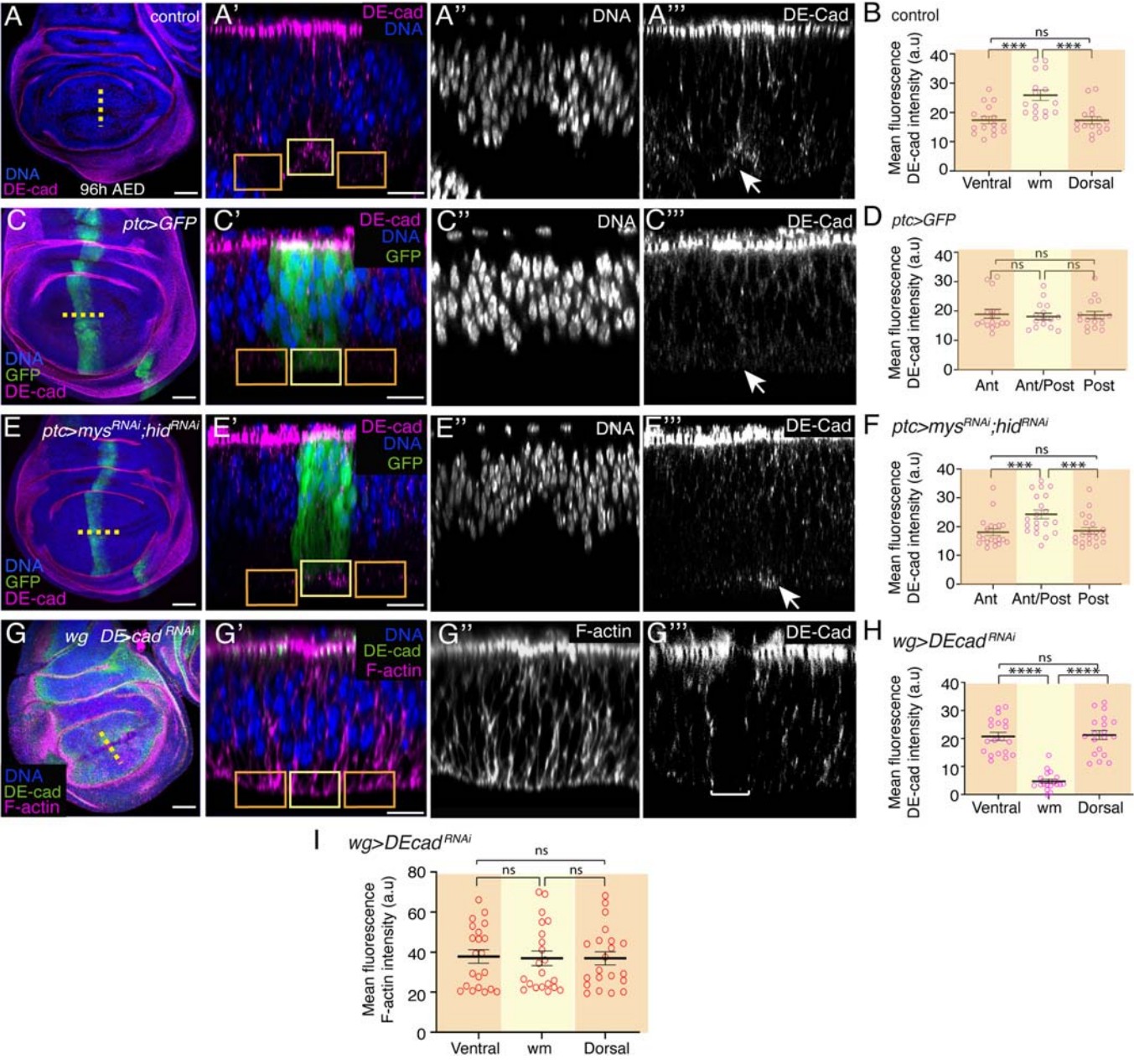

**Figure 5. Integrins regulate DE-cad localization.**

(A–G''') Confocal views of 96 h AED third-instar wing discs of the designated genotypes, stained with anti-DE Cad (magenta in **A, A', C, C', E, E'**, green in **G, G'** and white in **A''', C''', E''', G'''**), the nuclear marker Hoechst (DNA, blue in **A, A', C, C', E, E', G, G'** and white in **A'', C'', E''**), anti-GFP (green in **C, C', E, E'**) and Rhodamine Phalloidin to detect F-actin (magenta in **G, G'** and white in **G''**). Confocal YZ (**A'–A'''**, **G'–G'''**) and XZ (**C'–C'''**, **E'–E'''**) cross-sections along the yellow dotted lines shown in **A, C, E** and **G**, respectively. White arrows in (**A''', C''', E'''**) and bracket in **G'''** point to the wing margin region. (**B, D, F, H**) Quantification of anti-DE-Cad levels in controls and experimental wing discs in the regions framed in (**A'**), (**C'**), (**E'**) and (**G'**) (orange and yellow boxes). Multiple Mann–Whitney U test from left to right: (**B**) ***$p = 0.00078$, ***$p = 0.00045$, ns not significant, (**D**) ns not significant, (**F**) ***$p = 0.00024$, ***$p = 0.00072$, ns not significant, (**H**) ****$p = 2.4\text{e}{-5}$, ****$p = 4.8\text{e}{-6}$, ns not significant. Error bars represent the mean ± SEM. At least 15 wing discs were assessed over three independent experiments. Scale bar in all panels, 30 µm. Source data are available online for this figure.

the model by simulating this experimental situation by reducing the integrin adhesion strength without inducing basolateral actomyosin contractility. The model predicted that this would be unable to trigger any local indentation of the tissue at its basal side (Fig. 6A, Movie EV3). We next tested the model prediction by blocking F-actin accumulation in wing margin cells through the expression

of RNAis against components of the WAVE regulatory complex (WRC), known to control actin cytoskeletal dynamics, such as the Abelson interacting protein (Abi, Fig. 6D,E,H), which has been shown to cause a reduction in F-actin levels in *Drosophila* follicle cells (Cetera et al, 2014), or Scar/WAVE (Fig. 6F–H). We found that the expression of RNAis against any of these components of

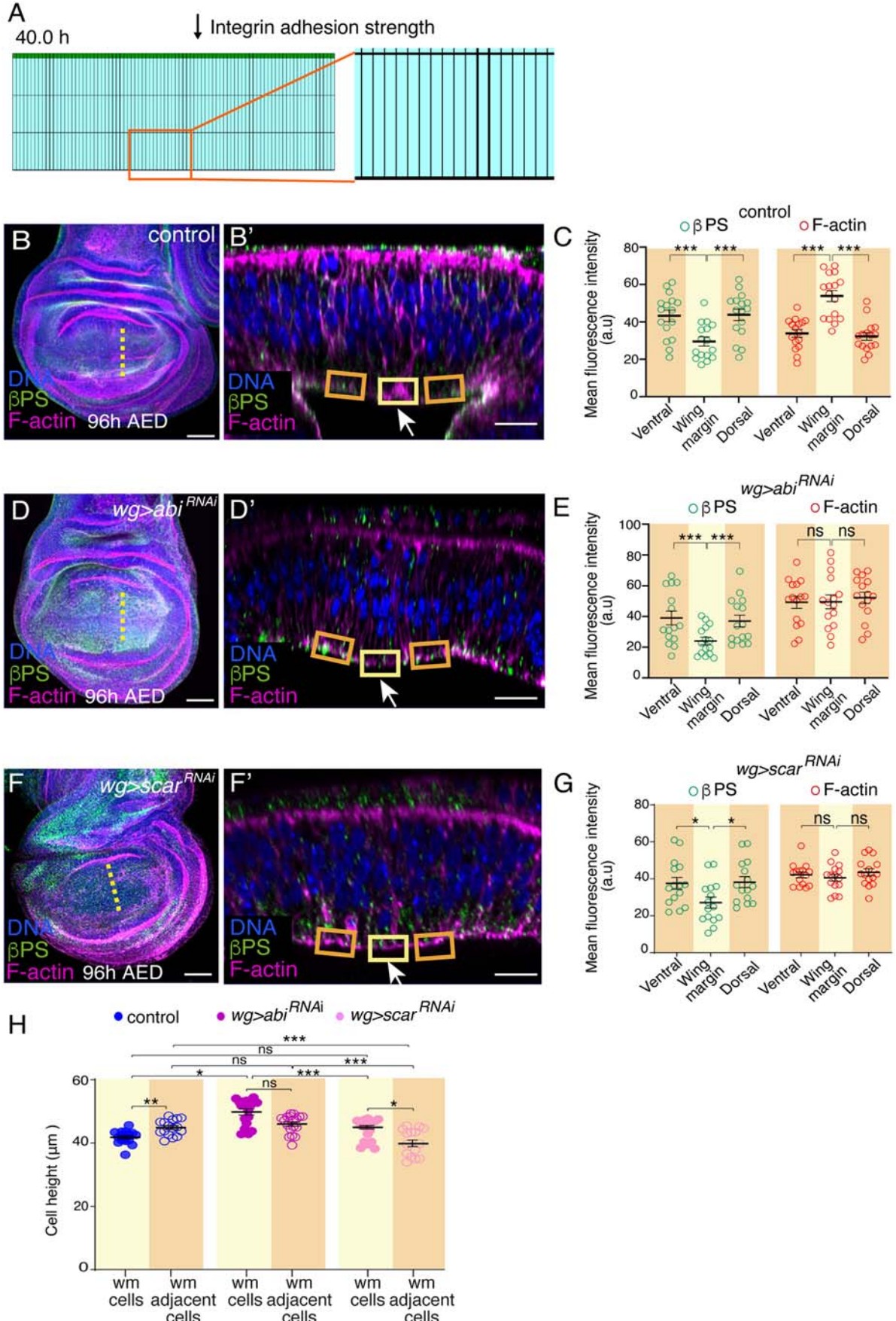

**Figure 6.  Downregulation of F-actin levels abolishes basolateral accumulation of F-actin and cell shortening in the wing margin.**

(A) Snapshot of a simulation where the integrin adhesion strength was reduced without inducing basolateral contractility. (B–B', D–D', F–F') Confocal views of wing imaginal discs of the indicated genotypes stained with anti-βPS (green), Rhodamine Phalloidin to detect F-actin (magenta) and the nuclear marker Hoechst (DNA, blue). (B, D, F) Maximal projections of control (B) and wing discs expressing an $abi^{RNAi}$ (D) or a $scar^{RNAi}$ (F) under the control of $wgGal4$. (B', D', F') Confocal YZ cross-sections along the yellow dotted lines shown in B, D and F. (C, E, G) Quantification of βPS and F-actin levels in control (C), $wg>abi^{RNAi}$ (E) and $wg>scar^{RNAi}$ (G) wing discs in the regions framed in (B'), (D') and (F'). Multiple Mann–Whitney U test from left to right: (C) $***p = 0.000138$, $***p = 0.00098$, $***p = 0.00069$, $***p = 0.0002$, (E) $***p = 0.00076$, $***p = 0.00085$, ns not significant, (G) $*p = 0.016$, $*p = 0.015$, ns not significant. Error bars represent the mean ± SEM. (H) Quantification of the height of wing margin and adjacent cells in control, $wg>abi^{RNAi}$ and $wg>scar^{RNAi}$ wing discs. Multiple Mann–Whitney U test from left to right: $**p = 0.00186$, $*p = 0.049$, $***p = 0.00023$, $***p = 0.00038$, $***p = 0.00049$, $*p = 0.02606$, ns not significant. Error bars represent the mean ± SEM. Scale bar in all panels, 30 μm. At least 15 wing discs were assessed over three independent experiments. Source data are available online for this figure.

the WRC in wing margin cells, by means of the $wgGal4$ line ($wg>abi^{RNAi}$ or $wg>scar^{RNAi}$), caused a general reduction in F-actin levels, including the accumulation on the basolateral side (Fig. 6B–G), cell shortening (Fig. 6H) and local basal tissue indentation (Fig. 6B',D',F'), despite the fact that integrin levels were still downregulated (Fig. 6B–G). Similarly, we found that reducing MyoII activity in wing margin cells, by expression of an RNAi against Rho 1 ($wg>Rho\ 1^{RNAi}$), was also able to block basal tissue indentation (Appendix Fig. S3A–C').

This result supports the view that basolateral actomyosin accumulation is most likely necessary to initiate basal folding.

### Downregulation of integrin levels is necessary for the initiation of basal folding

To analyze the contribution of reduced integrin adhesion strength to the initiation of basal folding, we ectopically maintained high levels of integrins in wing margin cells. To do this, we used the $wgG4$ line to overexpress in wing margin cells a constitutively active version of the PS2 integrin, which contains the βPS subunit and an αPS2 subunit carrying a deletion of its cytoplasmic domain ($wgG4>\alpha PS2\Delta cyt;\beta PS$; Martin-Bermudo et al, 1998). We found that the overexpression of this activated integrin in wing margin cells prevented the basolateral accumulation of F-actin (Fig. 7A,B'',C,D) and p-Sqh (Fig. EV4B–H), reorganization of basal F-actin (Fig. 7A''',B'''), cell shortening, the formation of a local basal indentation (Fig. 7B'',E) and the apically-biased nuclear positioning (Fig. EV2C''',D). Accordingly, our simulation showed that preventing integrin adhesion weakening and basolateral actomyosin accumulation hampered folding (Fig. 7F, Movie EV4).

Integrins are, in many aspects, essential for cell-fate determination (reviewed in (Streuli, 2009). To test whether the change in the behavior of wing margin cells overexpressing activated integrins could be due to a defect in proper cell fate acquisition, we analyzed the expression of the determinant of wing margin fate, $wg$, in $wg>\alpha PS2\Delta Cyt;\beta PS$ cells. We found that the normal pattern of Wg expression was not affected in $wg>\alpha PS2\Delta Cyt;\beta PS$ wing discs (Appendix Fig. S4A–C').

The loss of basolateral actomyosin accumulation in $wg>\alpha PS2\Delta Cyt;\beta PS$ wing discs prevented us from analyzing experimentally the consequences of reducing integrin levels without affecting basolateral actomyosin accumulation. Therefore, we used our computational model to test this scenario. We found that an increase in basolateral contractility in our computational model, without changing integrin adhesion strength, resulted in a small folding (Fig. 7G, Movie EV5). However, unlike in the real situation, we found that both sides of the integrin adhesion layer deformed, as

cells could not detach from the BM. This suggests that even though an increase in basolateral contractility could somehow initiate folding, downregulation of integrin adhesion strength is a prerequisite for proper and complete folding.

Altogether, these results suggest that we need the combined action of reducing adhesion strength and increasing basolateral contraction to trigger proper initiation of basal folding.

### Disrupting the initial steps of basal disc folding occurring at larval stages affects proper basal folding at pupal stages and wing morphogenesis

To analyze the morphogenetic consequences of disrupting the initial steps of basal disc folding on the proper folding of the wing disc, we analyzed wing discs at white pupae stage (2 h AP). We found that when we prevented either the increase in basolateral contractility ($wg>abi^{RNAi}$, Fig. 8C–C') or the reduction in integrin adhesion strength (wg > αPS2ΔCyt; βPS, Fig. 8E,E') in wing margin cells, bending along the wing margin was prevented (Fig. 8A',C',E'). Instead, the wing disc bent at random positions, leading to defective facing of the dorsal and ventral surfaces. The resulting adult wings showed a bubble phenotype (Fig. 8B,D,F), possibly as a consequence of faulty apposition of the ventral and dorsal surfaces. In contrast, ectopic reduction of integrin levels ($ptc>mys^{RNAi};hid^{RNAi}$) resulted in the formation of an ectopic indentation (Fig. 8G–I') and the appearance of an ectopic fold in the adult wing (Fig. 8H,J).

These results demonstrate that the reduction in integrin levels and the increase in basolateral contractility we observed during the initial steps of basal folding at L3 are necessary for the proper basal folding of the tissue during pupariation and for accurate wing formation.

## Discussion

Folding of epithelial sheets, either on their apical or basal surface, into complex 3D structures is a main driver of morphogenesis. This process requires precise and coordinated cell shape changes. These are driven by forces generated by the contraction of actomyosin networks, which are linked to the cell membranes by means of adherens or integrin-based junctions (reviewed in (Clarke and Martin, 2021). However, while progress has been made towards understanding apical folding, little is known about mechanisms underlying basal folding. Here, using the *Drosophila* wing disc epithelium as model system, we show that, in contrast to what happens on the apical side, where adherens junctions are essential for apical folding, adhesion to the ECM needs to be downregulated locally to allow basal folding.

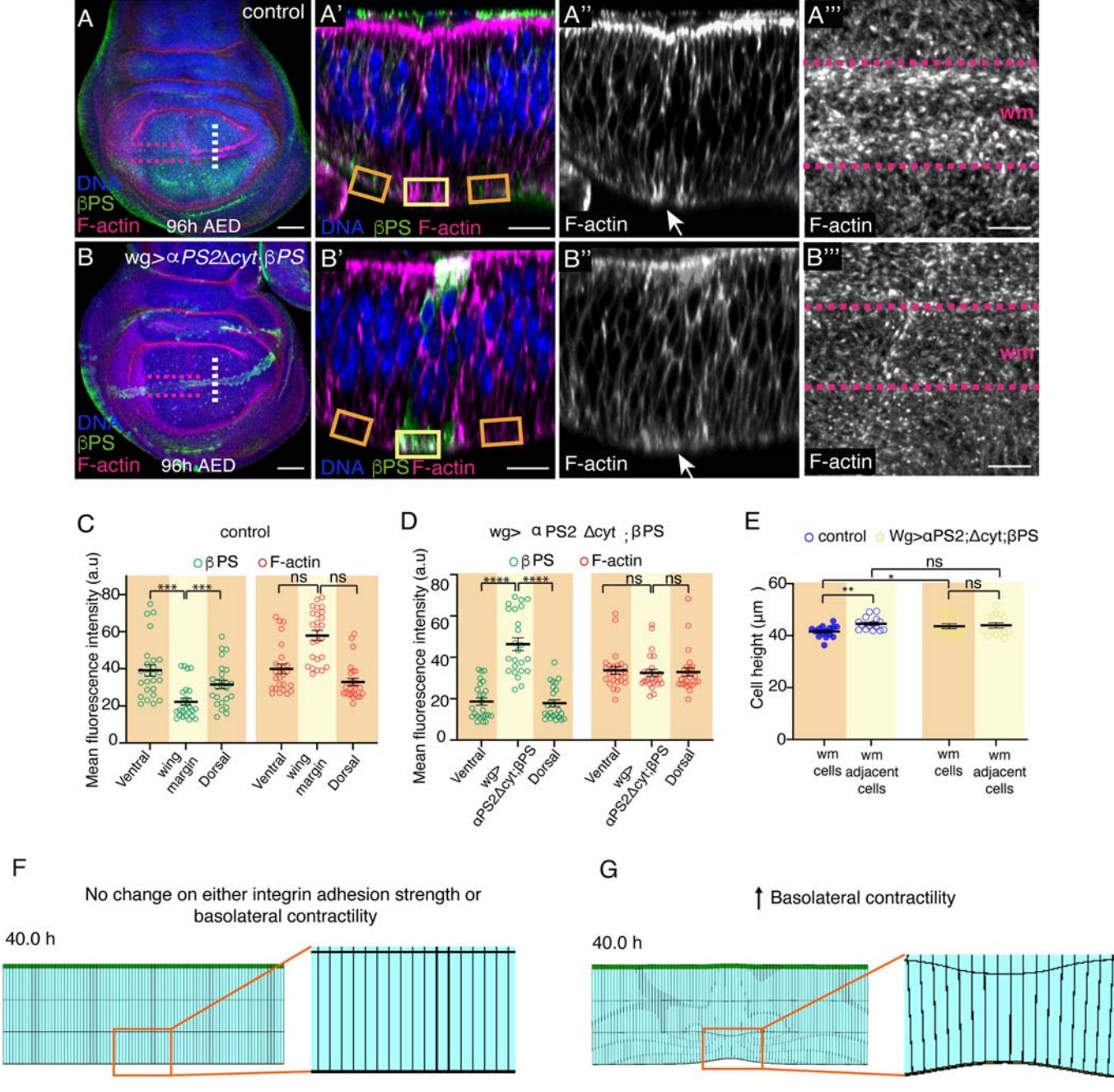

**Figure 7. Maintenance of high levels of integrins in the wing margin blocks actin reorganization and cell shortening.**

(A–B''') Confocal views of third-instar wing imaginal discs stained with anti-βPS (green), the F-actin marker Rhodamine Phalloidin (magenta in **A**, **A'**, **B**, **B'**, white in **A''**, **A'''**, **B''**, **B'''**) and the nuclear marker Hoechst (DNA, blue). (**A**) Control wing disc. (**B**) Wing disc co-expressing an active form of the αPS2 subunit (*αPS2ΔCyt*) and the βPS subunit under the control of *wgGal4* (*wg>αPS2ΔCyt; βPS*). (**A'**, **A''**, **B'**, **B''**) Confocal YZ cross-sections taken along the white dotted lines shown in (**A**, **B**). (**A'''**, **B'''**) Super-resolution images of XY sections taken in the region between the magenta dotted line in (**A**, **B**). (**C**, **D**) Quantification of βPS and F-actin levels in control (**C**) and *wg>αPS2ΔCyt; βPS* (**D**) wing discs. Multiple Mann–Whitney U test from left to right: (**C**) ***$p = 0.00011$, ***$p = 0.000434$, ns not significant, (**D**) ****$p = 3.1e{-}9$, ****$p = 2,7e{-}9$, ns not significant. Error bars represent the mean ± SEM. (**E**) Quantification of the height of wing margin and adjacent cells in control and *wg> αPS2ΔCyt; βPS* wing discs. Multiple Mann–Whitney U test from left to right: **$p = 0.01$, *$p = 0.0478$, ns not significant. Error bars represent the mean ± SEM. Scale bar in all panels, 30 μm. At least 15 wing discs were assessed over three independent experiments. (**F**) Snapshot of a simulation where no integrin adhesion weakening or basolateral contractility was applied. (**G**) Snapshot of a simulation where only basolateral contractility was applied, without changing integrin adhesion strength. Magnifications of regions framed in the snapshots in (**F**) and (**G**) are also shown. Source data are available online for this figure.

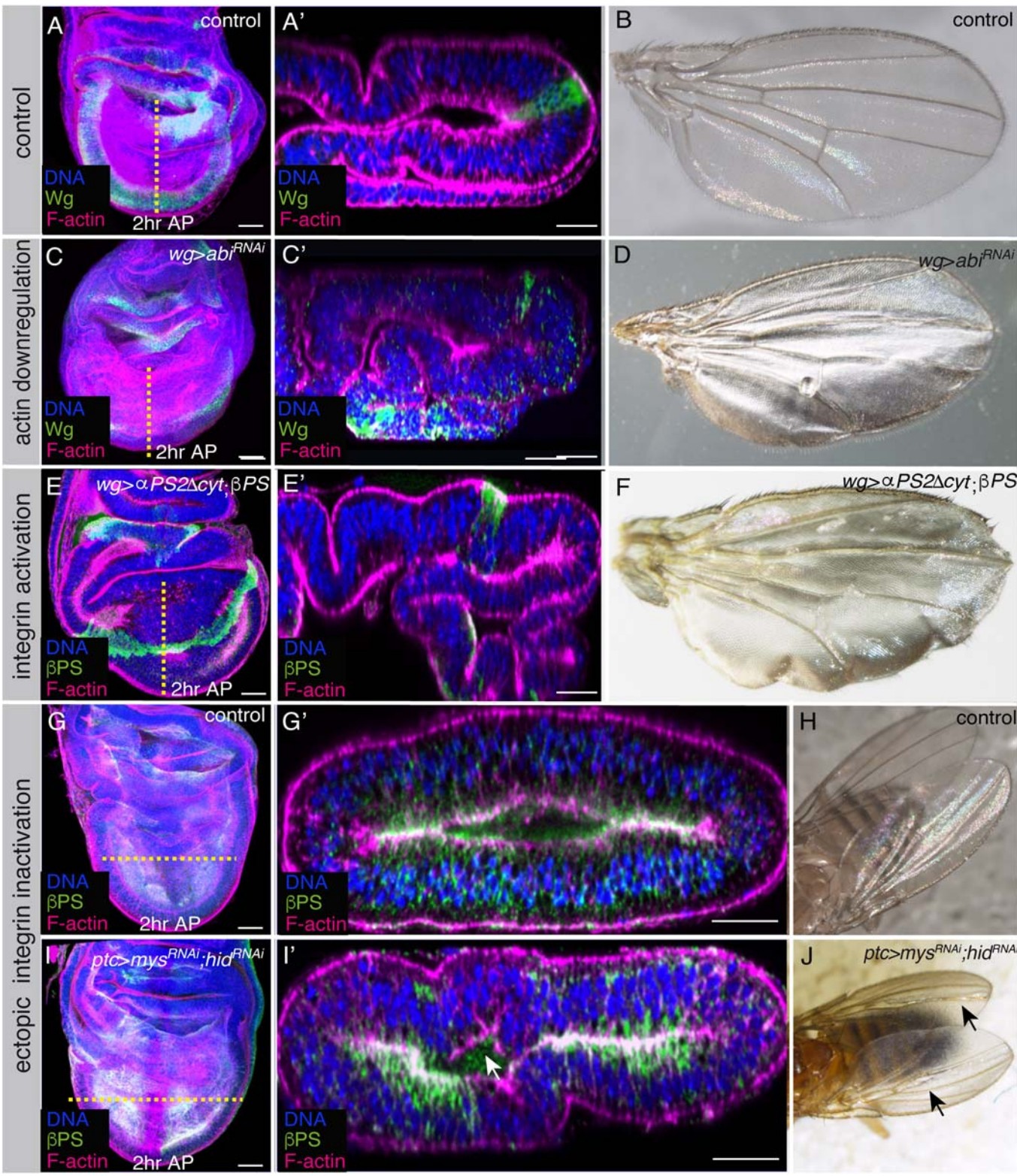

**Figure 8.   Manipulation of integrin levels causes defects in wing disc folding and in the adult wing.**

(A, C, E, G, I) Confocal views of 2 h APF wing imaginal discs of the indicated genotypes, stained with anti-βPS (green in E, E', G, G', I, I'), Rhodamine Phalloidin to detect F-actin (magenta) and the nuclear marker Hoechst (DNA, blue). (A, C, E) Maximal projections of control (A) and wing discs expressing an $abi^{RNAi}$ (C, $wg>abi^{RNAi}$) or co-expressing an active form of the αPS2 subunit and the βPS subunit (E, $wg>αPS2ΔCyt; βPS$) under the control of wgGal4. (A', C', E') Confocal cross-sections along the yellow dotted lines shown in (A, C, E). (B, D, F) Images of control (B), $wg>abi^{RNAi}$ (D) and $wg>αPS2ΔCyt; βPS$ (F) adult wings. (G, I) Maximal projections of control (G) and wing discs co-expressing RNAis against mys and hid under the control of the ptcGal4, $ptc>mys^{RNAi};hid^{RNAi}$ (I). (G', I') Confocal cross-sections along the yellow dotted line shown in (G, I). (H, J) Images of control (H) and $ptc>mys^{RNAi};hid^{RNAi}$ (J) adult wings. Scale bar in all panels, 30 μm. Source data are available online for this figure.

Actomyosin networks are assembled in distinct regions within the cell where they exhibit distinct architectures. On the apical side of epithelial cells two cortical networks can be found: a cortical two-dimensional network located in the middle region below the plasma membrane (referred to as 'medial' or 'medioapical') and/or a circumferential actomyosin belt underlying adherens junctions (Martin and Goldstein, 2014). Both can drive apical constriction (Ishiuchi and Takeichi, 2011; Lecuit et al, 2011; Salbreux et al, 2012; Wu et al, 2014). In contrast, the most frequent actomyosin network found on the basal side of epithelial cells are stress fibers, which, by exerting traction forces on the external environment and generating isometric tension, normally help cells retain their shape (Kumar et al, 2006). Nonetheless, stress fibers have also been involved in cell contraction processes in *Drosophila* follicle cells, where periodic pulses of actomyosin bundles lead to oscillations of the basal surface. However, this process does not culminate in an enduring reduction of the basal surface (He et al, 2010). Hence, how do cells constrict basally to allow basal epithelial folding? Here, we show that, during the basal bending of the wing disc along the wing margin, a basolateral reorganization of the actomyosin network occurs specifically in wing margin cells. Actomyosin networks have been shown not to be static. Furthermore, transitions between them have been proposed to drive several processes underlying morphogenesis (reviewed in (Chalut and Paluch, 2016). How these transitions are controlled remain elusive. Here, we demonstrate that reducing adhesion to the BM mediated by integrins is necessary and sufficient to induce a reorganization of the basal actomyosin network in wing margin cells and basal surface reduction. This result suggests that integrins block transitions between basal actin networks, most likely to maintain tissue shape. This is in agreement with our previous results showing that elimination of integrins from follicle cells leads to a reorganization of F-actin from stress fibers into a dynamic cortical distribution. Furthermore, similar to what happens in wing margin cells, integrin reduction in follicle cells also causes a shrinkage of the basal surface, although, in this case, this results in tissue architecture disruption (Santa-Cruz Mateos et al, 2020).

A recent study has demonstrated the existence of basal spot junctions containing E-cad and Hippo pathway proteins in *Drosophila* epithelial tissues (Kroeger et al, 2024). Here, we show an enrichment of basal DE-Cad levels specifically in wing margin cells. In addition, we found that the local ectopic reduction of integrin levels is sufficient to cause an increase in basal DE-Cad levels. Furthermore, we show that cadherins are necessary for the changes in actin dynamics and cell shape underlying the initiation of basal folding. Cadherins and integrins are linked through the actin cytoskeleton. Furthermore, mechanically driven crosstalk between them regulates not only their spatial distribution but also the spatial organization of the actin cytoskeleton, molecular signals

and forces, all of which guide processes from cell fate determination to ECM patterning and cell migration (reviewed in (Mui et al, 2016). Our results suggest that the modulation of the mechanical landscape of the cell through the crosstalk between integrins and cadherins is essential for the regulation of epithelial folding. Finally, our results also show that the downregulation of integrin levels induces a basal enrichment of the MyoII activator Rok. Altogether, our results support a model for basal folding in which the basal side of the cells would appropriate the players used for apical folding, an active actomyosin network, DE-Cad and Rok, to induce the needed cell shape changes. This requires and is triggered by a down-regulation of cell-ECM interactions (Fig. 9). In contrast to what we show here, in *Drosophila* follicle cells, cell-matrix adhesion has been shown to positively control basal Rho1-MyoII activity and dynamics (Qin et al, 2017). In this case, the actomyosin cytoskeleton is organized in stress fibers that undergo periodic oscillations. Thus, we propose that actomyosin regulators interact with each other in different ways depending on the cellular context, the actomyosin network architecture and the desired outcome of the contractile response.

As adhesion sites and contractile actomyosin activity are coupled during morphogenesis (reviewed in Heisenberg and Bellaiche, 2013), the strong interdependence between actomyosin activity and cell adhesion hampers the evaluation of the individual contribution and the temporal hierarchy of action of these factors during folding. Computational modeling has emerged as a useful tool to address these types of questions. Our simulations show that a reduction in integrin levels is not sufficient on its own to account for the cell shape changes triggering basal folding along the wing margin. In contrast, local actomyosin accumulation leads to a significant evagination of the tissue. However, in this case, the shape of the fold does not completely recapitulate the observed experimental folding. This suggests that even though actomyosin activity might be the driving force for local basal epithelial folding, detachment from the BM is necessary for this process to occur properly. Our model also predicts that the events that take place at the wing margin prior to basal folding have to follow a precise temporal order. Thus, our model forecasts that the reduction in cell-BM adhesion has to precede basolateral actomyosin accumulation. This is in contrast to what has been computationally proposed for the process of bending that the whole wing pouch undergoes to achieve its stereotypical dome shape. In this case, the role of the BM was predicted to follow the generation of a bent profile by actomyosin contractility and served just to maintain tissue bending (Nematbakhsh et al, 2017). The difference between this observation and ours may arise from the extent of epithelial bending being studied, global versus local bending. In our case, asymmetry needs to be broken in a group of cells to allow local changes in cell shape and bending along a narrow region, while in the case of the wing

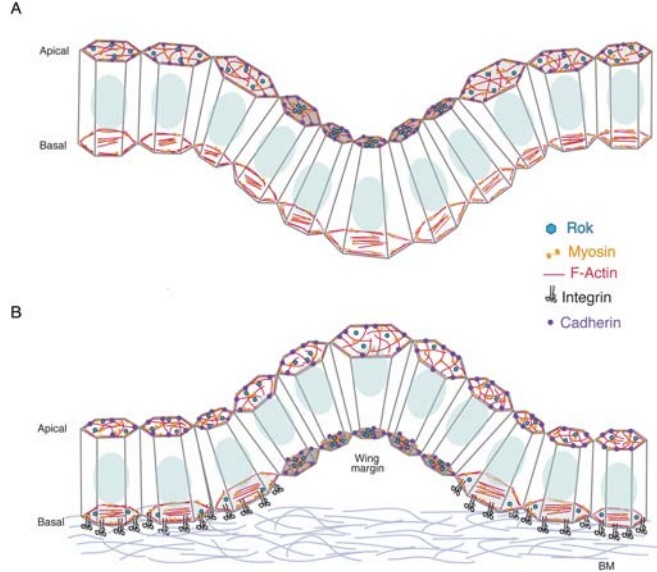

**Figure 9. Model illustrating the mechanisms of basal versus apical folding.**

(A) During apical folding, apical constriction mediated by forces exerted by the cortical actomyosin network and the transmission of these forces to cadherin-based adherens junctions initiates folding. (B) On the basal side, folding is initiated by a reduction on integrin-mediated adhesion to the BM that triggers a basolateral reorganization of the actin cytoskeleton, the basal localization of cadherins and Rok and basal constriction.

pouch the whole tissue needs to curve. This further supports the idea that the mechanical mechanisms that epithelia have adopted to generate folds, either local or on a more global tissue-wide scale, are many and can be specifically combined to unambiguously shape the various epithelial tissues.

While folding of epithelial tissues towards the apical surface has long been studied, the molecular and cellular mechanisms that mediate epithelial folding towards the basal surface are just emerging. Besides *Drosophila* wing development, basal tissue folding is required for different morphogenetic events in both vertebrate and invertebrate systems, such as the formation of the midbrain-hindbrain boundary (Gutzman et al, 2008) and the optic cup in zebrafish (Martinez-Morales et al, 2009), *Ciona* notochord elongation (Dong et al, 2011) or Hydra bud formation (Holz et al, 2017). Discovering new mechanisms of epithelia folding is crucial to better understand epithelia shaping during normal and developmental disease conditions, as well as for the future of tissue engineering and regenerative medicine.

## Methods

### Reagents and tools table

| Reagent/Resource | Reference or Source | Identifier or Catalog Number |
|---|---|---|
| **Experimental models** | | |
| *UAS-mys^RNAi* | Bloomington Drosophila Stock Center | N/A |

| Reagent/Resource | Reference or Source | Identifier or Catalog Number |
|---|---|---|
| *UAS⁻hid^RNAi* | Bloomington Drosophila Stock Center | 13957 |
| *UAS-abi ^RNAi* | DGRC-Kyoto | 9749R |
| *UAS-scar ^RNAi* | Bloomington Drosophila Stock Center | 4223 |
| *UAS-Abi RNAi* | DGRC Kyoto | 9749R |
| *UAS-αPS2_{ΔCyt};UAS-βPS* | (Martin-Bermudo et al, 1997) | N/A |
| *wg Gal4* | Bloomington Drosophila Stock Center | 63048 |
| *ptc Gal4* | Bloomington Drosophila Stock Center | 8915 |
| *UAS-CD8GFP* | Bloomington Drosophila Stock Center | 79626 |
| mNGRok | Kindly provided by *Dr Katja Roeper* | N/A |
| ResilleGFP | Kindly provided by Prof. Wieschaus | N/A |
| **Recombinant DNA** | | |
| **Antibodies** | | |
| Rabbit anti-pMyo light chain 2 | Cell Signaling | 3671S |
| Goat anti- GFP^FICT | Abcam | Ab6673 |
| Mouse anti-βPS | DSHB | CF6G111 |
| Rat anti-DE-Cad | DSHB | AB528120 |
| Mouse anti-wingless | DSHB | EPR3189 |
| Anti-Rabbit Cy3 | Jackson ImmunoReseach Laboratories, Inc | 211-502-171 |
| Anti-Rabbit Cy5 | Jackson ImmunoReseach Laboratories, Inc | 211-502-172 |
| **Oligonucleotides and other sequence-based reagents** | | |
| **Chemicals, Enzymes and other reagents** | | |
| Rhodamine-Phalloidin | Molecular Probes | R37112 |
| Hoechst | Thermo Fisher | H21491 |
| **Software** | | |
| **Other** | | |

## Methods and protocols

### Drosophila husbandry

*Drosophila melanogaster* strains were raised at 25 °C on standard medium. To knock down integrins, *hid, abi, scar, DE-cad* or to increase integrin activity in wing disc cells, the *patched* (*ptc*, a gift from Dra. I. Guerrero) and the *wingless* (*wg* a gift from Prof. F. Casares) Gal4 drivers were used in combination with the following lines: *UAS-mys^RNAi*, *UAS⁻hid^RNAi*, *UAS-abi^RNAi* (DGRC-Kyoto 9749 R), *UAS-scar^RNAi*, *UAS-DE-cad^RNAi* and *UASαPS2ΔCyt;UASβPS* (Martin-Bermudo et al, 1997). To visualize cell membranes in vivo, the membrane markers Resille-GFP and CD8GFP were used. Rok localization was observed using the mNGRok marker (a gift from Dr Katja Roeper).

## Immunohistochemistry and imaging

Wing imaginal discs were stained using standard procedures and mounted in Vectashield (Vector Laboratories, Burlingame, CA, United States). The following primary antibodies were used: goat anti-GFP[FICT] (Abcam, 1:500), mouse anti-βPS, rat anti-DE-Cad (DCAD2) and anti-wingless (DSHB, University of Iowa, USA, 1:50), rabbit anti-pMyosin light chain 2, p-Sqh (Cell Signalling 1:20). The secondary antibodies used were Alexa fluor 488 (Molecular Probes[TM]) and Cy3 and Cy5 (Jackson ImmunoReseach Laboratories, Inc.) at 1:200. For actin labeling, fixed wing imaginal discs were incubated with Rhodamine Phalloidin (Molecular Probes, 1:40) for 40 min. DNA was labeled using Hoechst (Molecular Probes, 1:1000). xyz images were maximum projections of a z-stack (typically between 20 and 30 steps with a step size of 1 μm). Cross-sections were single sections collected at the focal plane indicated in the figures except for the images at 2 h AP, which were orthogonal views of a z-stack of the entire wing disc (between 250 and 300 steps with a step size of 0.6 μm). Pictures were processed with the Adobe Photoshop CS6 software. Confocal images were obtained using Leica SP5-MP-AOBS and Stellaris confocal microscopes, equipped with a Plan-Apochromat 40X oil objective (NA 1.4). Super-resolution images were obtained using a Zeiss LSM 880 with Airyscan, equipped with a Plan-Apochromat 63X oil objective. No less than 13 wing imaginal discs were analyzed per genotype.

## Quantification of samples

To quantify fluorescent intensity of the different markers in confocal xzy views, regions of interest were outlined manually and mean fluorescent intensity was measured using FIJI-ImageJ software (Schindelin et al, 2012). Measurements represented in the graph correspond to the average mean of fluorescence intensity in the basolateral region, normally represented in the figures by a rectangle, where the height of the rectangle represents the portion of the lateral side of the cells that has been measured, up to 7 μm from the most basal side of the cells. To quantify fluorescent intensity in confocal xyz views, maximal projections of 3–5 confocal stacks were produced. Regions of interest were selected manually and mean fluorescent intensity was measured using FIJI-ImageJ software.

To calculate cell height, a vertical line was drawn from apical to basal surface in the regions of interest in the epithelium, using F-actin staining as a reference. The total length of the resulting line was measured using FIJI-ImageJ software.

To measure wing margin width in cross-sections, wg>GFP signal was used as a reference. A horizontal line was drown occupying the entire width of wg>GFP signal, in the apical (AL width) and in the basal side (BL width) of the wing epithelium. The total length of the resulting line was measured using FIJI-ImageJ software.

Calculation of fractional nuclear positioning was performed using the formula: Fractional positioning = $L_B/(L_B + L_A)$ where $L_B$ was the length from the center of the nucleus to the basal surface of the epithelium and $L_A$ the length from the center of the nucleus to the apical surface.

## Statistical analysis

Statistically significant differences between control and experimental samples were distributions and the Student´s t-test for normal distributions. The quantification plots show individual measurements, together with their mean and standard error. A minimum of 15 wing discs per condition was analyzed.

## Computational model

The Finite Element Model was adapted from a previously published model (Tozluoglu et al, 2019). The initial mesh was generated using custom-written code in Matlab. A cross-section of the columnar epithelium perpendicular to the DV boundary was modeled. The epithelium consisted of an apical layer and three cell body layers. Experimental measurements were used to define the height (47 μm, i.e., 45 μm of cell body layer plus 2 μm of actin layer) and DV length (130.5 μm) of the simulated tissue. The integrin adhesion layer was defined as a thin layer of 0.2 μm thickness. To account for the presence of the peripodial layer above the columnar epithelium, an external viscosity was applied to the apical side of the tissue. Wing margin region was defined as a region in the middle of the tissue along the DV boundary, accounting for ~10% of its total length. The parameters used in the model are as below. Equations were solved using PARDISO 6.0 Solver (Karipis and Kumar, 1999; Schenk and Gartner, 2004; Schenk and Gartner, 2006).

| Parameter | Description | Value/Range | Reference |
|---|---|---|---|
| Tissue dimensions | Length (DV length) | 130.5 μm | Measurements in this study |
| | Height | 47 μm, i.e., 45 μm of cell body layer plus 2 μm of actin layer | Measurements in this study; (Tozluoglu et al, 2019) |
| | Width | 45 μm | Arbitrary |
| Tissue viscoelasticity | | | |
| $E_{cells}$ | Young's modulus of the cell body layers | 25 Pa | (Tozluoglu et al, 2019) |
| $E_{actin}$ | Young's modulus of the actin-rich apical layer | 100 Pa | (Tozluoglu et al, 2019) |
| $\eta_{ext}$ | External viscous resistance coefficient | 10 Pa.s.μm$^{-1}$ (basal) 16,000 Pa.s.μm$^{-1}$ (apical) | (Tozluoglu et al, 2019) |
| $\upsilon$ | Poisson ratio of the tissue | 0.29 Pa.s.μm$^{-1}$ | (Schluck et al, 2013) |
| Integrin adhesion layer | | | |
| $E_{integrin}$ adhesion strength | Young's modulus of the integrin adhesion layer | 160 kPa (outside wing margin); 10$^{-6}$ Pa (wing margin) | Simulated for a range |
| Ideal height change | In conditions where basolateral contractility was imposed, the ideal height would change such that it would reach 60% of initial ideal height at 40 h simulation time. | 0 (outside wing margin) 0.015–0.01875 per hour (wing margin) | Simulated for a range |

## Data availability

This study includes no data deposited in external repositories.

The source data of this paper are collected in the following database record: biostudies:S-SCDT-10_1038-S44318-025-00384-6.

## Peer review information

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

## Acknowledgements

We thank the BDSC for reagents. Research in our laboratories is funded by the Spanish Ministerio de Economía y Competitividad and the FEDER programme (PID2022-143001NB-100 to MDM-B) and by the Junta de Andalucía (Proyecto de Excelencia P09-CVI-5058). We thank the National Centre for the Replacement, Refinement and Reduction of Animals in Research (NC3Rs) for funding this work (NC/T002425/1 and an Early Career Engagement Award to NK). NK was also supported by a Leverhulme Trust project grant (RPG-2020-068) awarded to YM. YM was funded by MRC award MR/W027437/1, a Lister Institute Research Prize and EMBO YIP.

## Author contributions

**Andrea Valencia-Expósito**: Conceptualization; Data curation; Formal analysis; Validation; Investigation; Visualization; Methodology; Writing—original draft; Writing—review and editing. **Nargess Khalilgharibi**: Conceptualization; Data curation; Formal analysis; Funding acquisition; Validation; Investigation; Visualization; Methodology; Writing—original draft. **Ana Martínez-Abarca Millán**: Data curation; Formal analysis; Validation; Investigation; Methodology; Writing—review and editing. **Yanlan Mao**: Conceptualization; Data curation; Software; Formal analysis; Supervision; Funding acquisition; Validation; Investigation; Visualization; Methodology; Writing—original draft; Writing—review and editing. **Maria D Martin-Bermudo**: Conceptualization; Resources; Data curation; Software; Formal analysis; Supervision; Funding acquisition; Validation; Investigation; Visualization; Methodology; Writing—original draft; Project administration; Writing—review and editing.

Source data underlying figure panels in this paper may have individual authorship assigned. Where available, figure panel/source data authorship is listed in the following database record: biostudies:S-SCDT-10_1038-S44318-025-00384-6.

## Disclosure and competing interests statement

The authors declare no competing interests.

# Expanded View Figures

**Figure EV1.  F-actin and Myosin organization changes throughout development in wing margin cells.**

(A–D) Confocal views of third-instar wing discs stained with anti-mys-GFP (green in **A–A′**, **B–B′**, **C–C′**, **D–D′** and white in **A″**, **B″**, **C″**, **D″**) and Rhodamine Phalloidin to detect F-actin (magenta in **A–A′**, **B–B′**, **C–C′**, **D–D′** and white in **A‴**, **B‴**, **C‴**, **D‴**). (**A, B, C, D**) Maximal projections of 80 h AED (**A, C**) and 96 h AED (**B, D**) wing discs. (**A′–A‴**, **B′–B‴**, **C′–C‴**, **D′–D‴**) High resolution images of YZ sections taken at the region in the dotted square in (**A, B, C, D**), respectively. (**E**) Quantification of *mys*-GFP, *sqh*-GFP and F-actin levels in mid L3 wing discs at the regions framed in (**A′**, **B′**, **C′**, **D′**). Multiple Mann–Whitney U test from left to right: ***$p = 0.0002$, ***$p = 0.0002$, ***$p = 0.0016$, ***$p = 0.0049$, ***$p = 0.0003$, ***$p = 0.0007$. Error bars represent the mean ± SEM. Scale bar in all panels, 30 µm. At least 15 wing discs were assessed over three independent experiments. Source data are available online for this figure.

▶

     

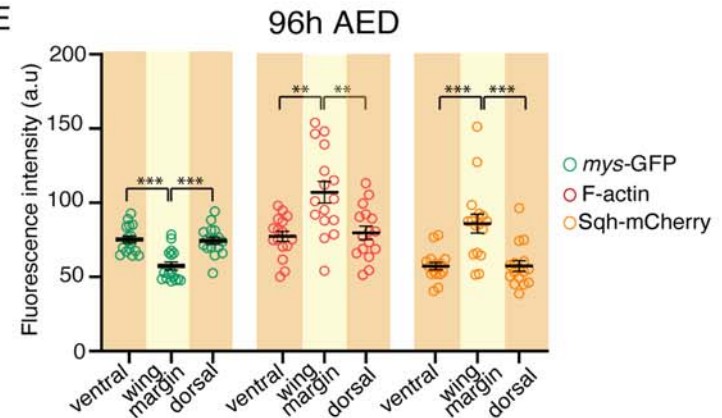

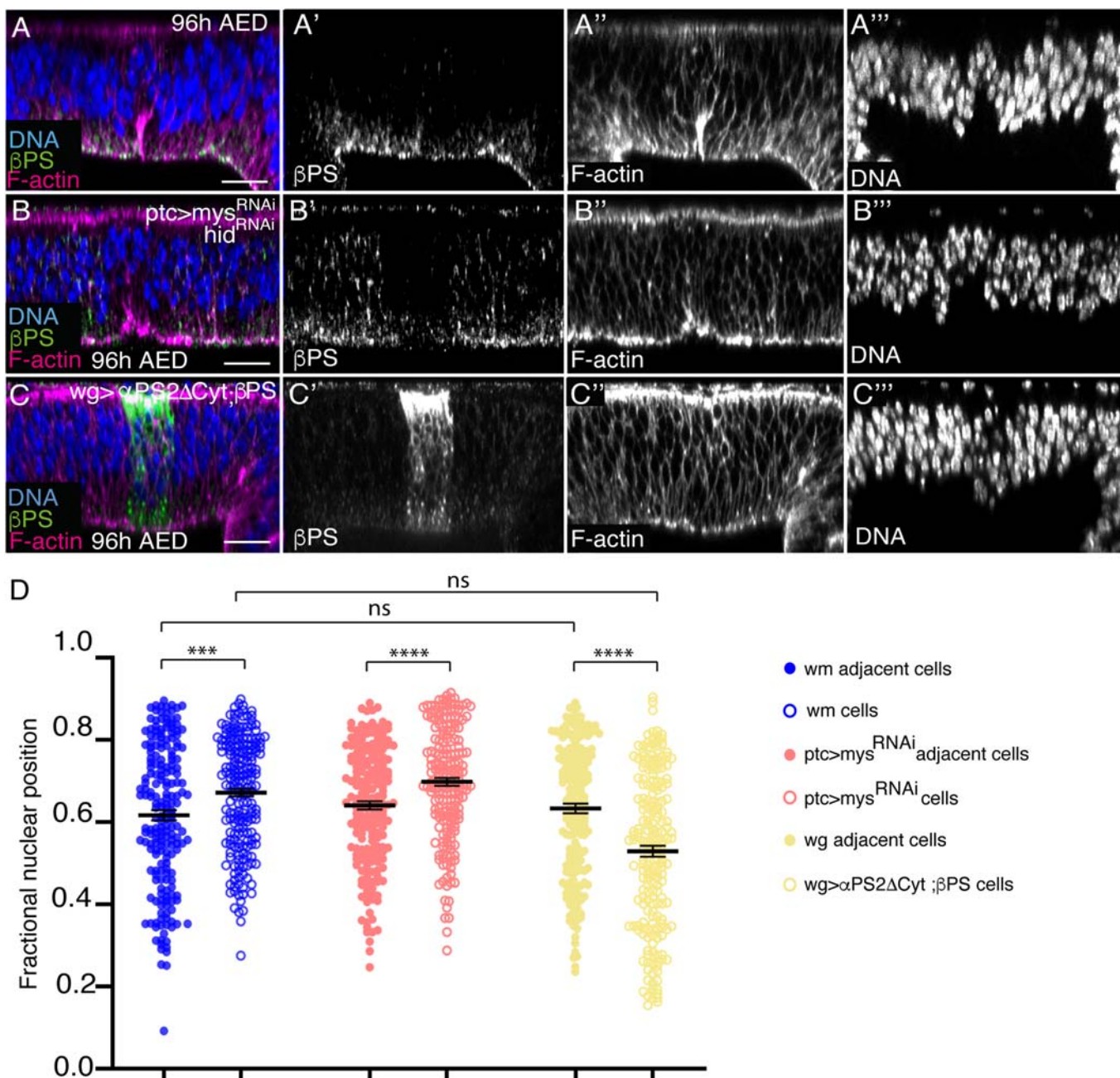

**Figure EV2. Integrin levels condition nuclear positioning.**

Confocal YZ (**A–A″**, **C–C″**) and XZ (**B–B″**) cross-sections of third-instar wing discs stained with anti-βPS (green in **A**, **B**, **C** and white in **A′**, **B′**, **C′**), Rhodamine Phalloidin to detect F-actin (magenta in **A**, **B**, **C** and white in **A″**, **B″**, **C″**) and the nuclear marker Hoechst (DNA, blue in **A**, **B**, **C** and white in **A‴**, **B‴**, **C‴**). (**A**) Control wing disc. (**B**) Wing disc co-expressing RNAis against *mys* and *hid* under the control of *ptcGal4, ptc>mys^{RNAi}; hid^{RNAi}*. (**C**) Wing disc co-expressing an active form of the αPS2 subunit and the βPS subunit under the control of *wgGal4, wg>αPS2ΔCyt; βPS*. (**D**) Quantification of the fractional nuclear position in wing margin and adjacent cells in control and experimental wing discs. Multiple Mann–Whitney U test from left to right: ****p* = 0.000455, *****p* = 1.62768E−05, *****p* = 7.05405E−09, ns not significant. Error bars represent the mean ± SEM. Scale bar in all panels, 30 μm. 200 nuclei were analyzed in at least 15 wing discs over three independent experiments. Source data are available online for this figure.

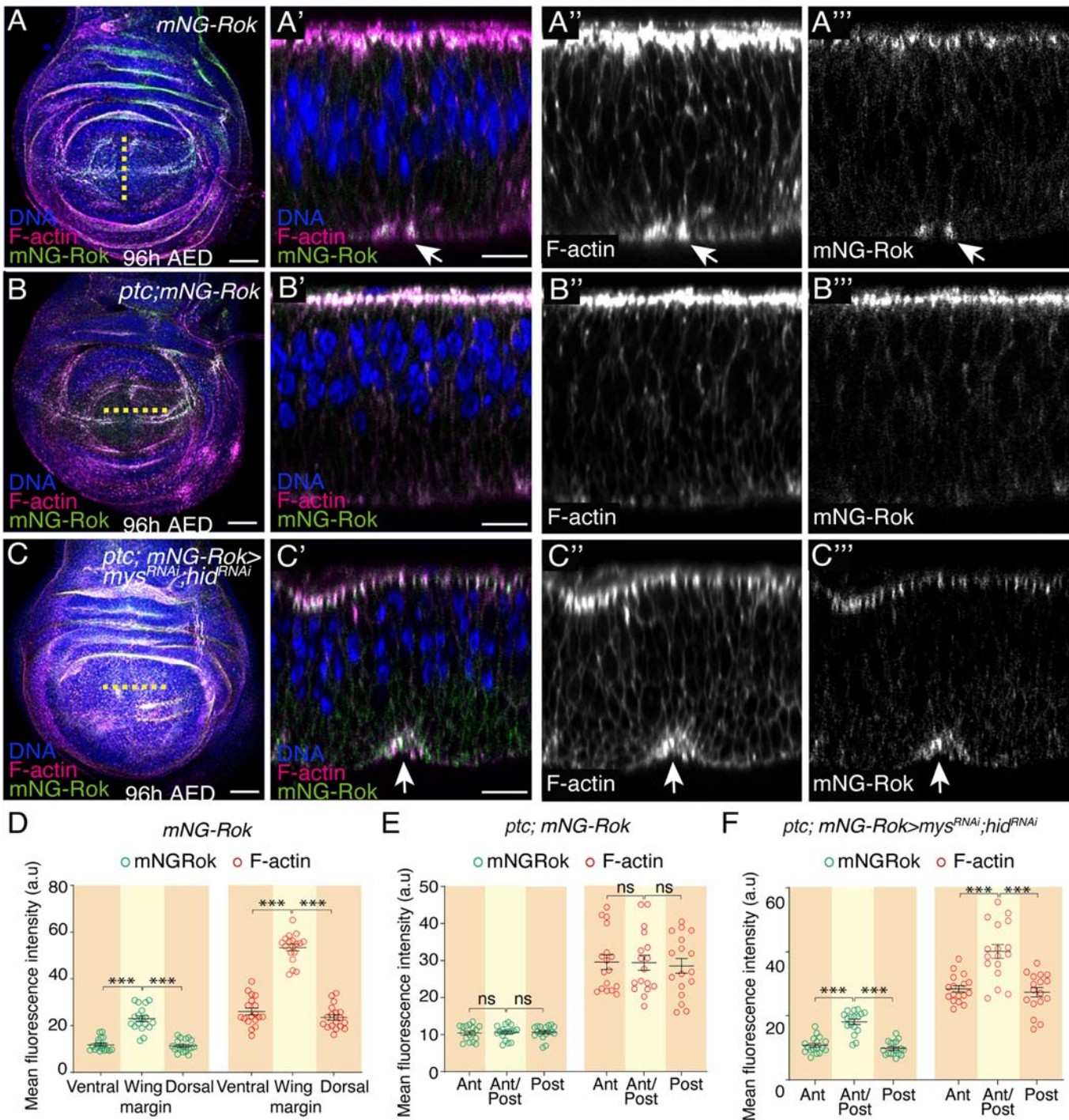

**Figure EV3. Rok is found basally in wing cells with low integrin levels.**

(A–C) Confocal views of third-instar wing discs of the indicated genotypes, stained with Rhodamine Phalloidin to detect F-actin (magenta **A**, **A'**, **B**, **B'**, **C**, **C'** and white in **A"**, **B"**, **C"**) and the nuclear marker Hoechst DNA (blue in **A**, **A'**, **B**, **B'**, **C**, **C'**). *mNGROK* signal is shown in green in **A**, **A'**, **B**, **B'**, **C**, **C'** and in white in **A"'**, **B"'**, **C"'** (**A–C**). Maximal projections of a third-instar wing imaginal discs of the indicated genotypes. Confocal YZ (**A'–A"'**) and XZ (**B'–B"'**, **C'–C"'**) cross-sections along the yellow dotted lines shown in (**A–C**). (D–F) Quantification of *mNGROK* levels in controls and experimental wing discs along the white dotted lines in (**A**), (**B**) and (**C**). Multiple Mann–Whitney U test from left to right: (**D**) ***$p = 0.002$, ***$p = 0.001$, ***$p = 0.0006$, ***$p = 0.00011$, (**E**) ns not significant, (**F**) ***$p = 0.0013$, ***$p = 0.00011$, ***$p = 0.00074$, ***$p = 0.00047$. Error bars represent the mean ± SEM. Scale bar in all panels, 30 μm. At least 15 wing discs were assessed over three independent experiments. Source data are available online for this figure.

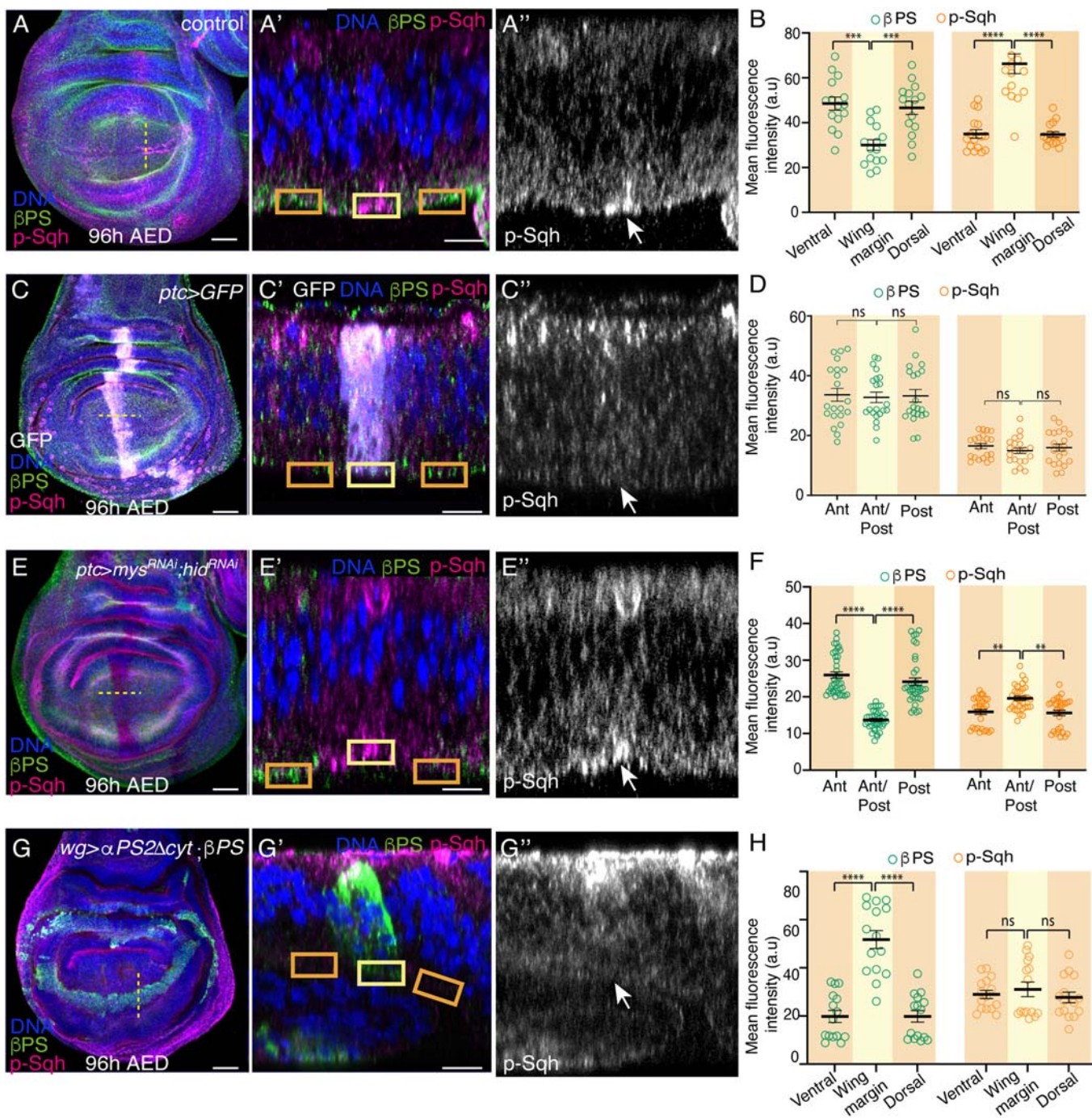

**Figure EV4.  Integrins regulate pSqh levels in wing margin cells.**

(A–A'', C–C'', E–E'', G–G'') Confocal views of third-instar wing discs of the designated genotypes stained with anti-βPS (green), anti-pSqh (magenta in **A, A', C, C', E, E', G, G'** and white in **A'', C'', E'', G''**), and the nuclear marker Hoechst DNA (blue in **A, A', C, C', E, E', G, G'**). (**A, C, E, G**) Maximal projections of a third-instar wing imaginal discs of the indicated genotypes. Confocal YZ (**A', A'', G', G''**) and XZ (**C', C'', E', E''**) cross-sections along the yellow dotted lines shown in (**A, C, E, G**). (**B, D, F, H**) Quantification of βPS and pSqh levels in control and experimental wing discs in the regions framed in (**A', C', E', G'**) (orange and yellow boxes). Multiple Mann–Whitney U test from left to right: (**B**) ***p = 0.00013, ***p = 0.0029, ****p = 1.1e−5, ****p = 1.1e−5, (**D**) ns not significant, (**F**) ****p = 4.4e−14, ****p = 6.1e−11, **p = 0.0024, **p = 0.0018, (**H**) ****p = 4.8e−14, ****p = 6.1e−11, ns not significant. Error bars represent the mean ± SEM. Scale bar in all panels, 30 μm. At least 15 wing discs were assessed over three independent experiments. Source data are available online for this figure.

