## [Peer Review File · The EMBO Journal]

Local weakening of cell-ECM adhesion triggers basal epithelia tissue folding

Andrea Valencia-Expósito, Nargess Khalilgharibi, Ana Martínez-Abarca Millán, Yanlan Mao, and Maria Dolores Martin-Bermudo

Corresponding authors: Maria Dolores Martin-Bermudo (mdmarber@upo.es) , Andrea Valencia-Expósito (avalexp@upo.es)

Review Timeline:

Transferred from Review Commons:	15th Nov 24
Editorial Decision:	21st Nov 24
Revision Received:	16th Dec 24
Editorial Decision:	23rd Jan 25
Revision Received:	30th Jan 25
Accepted:	3rd Feb 25

Editor: Ieva Gailite

Transaction Report:

This manuscript was transferred to The EMBO Journal following peer review at Review Commons.

Review #1**1. Evidence, reproducibility and clarity:****Evidence, reproducibility and clarity (Required)**

While infolding from the apical surface (as in the developing *Drosophila* embryo) has been examined previously, infolding from the basal surface is not as well understood and is unlikely to proceed through the same mechanism. How does it work? The authors take a strong approach to attacking this problem: they combine computational modeling with quantitative imaging and use a well-established and appropriate developmental system, the *Drosophila* imaginal wing disc. They show that folding at the basal side of the wing disc is modulated by both cell-matrix adhesion, which decreases locally, and cell-cell adhesion, which increases. The latter adhesion relies on pools of Shotgun (*Drosophila* E-Cadherin) that localize at the basal-most part of the lateral cortical domain. The data and interpretation are straightforward and convincing. I do not have concerns that require additional experimentation.

****Minor concerns:****

Red-green colorblind readers might struggle to interpret some of the confocal micrographs. It should be fairly easy to change the color scheme using Fiji (for example).

A trivial concern is that the Supplemental Figures weren't numbered. It's relevant to bring this up because in my opinion the model figure (I'm confident this is Sup. Fig. 8) would be a useful addition to the main article (space permitting).

Please break up the following sentence so that it's easier to read:

"In addition, we found that the distribution of activated non-muscle MyosinII, hereafter MyoII, as detected using an antibody that specifically recognizes the *Drosophila* homolog of the MyoII regulatory light chain, spaghetti squash (sqh), when it is phosphorylated at the activating Ser-21 (pSqh), followed a dynamic similar to that of F-actin, changing from a homogeneous distribution at early stages to a basolateral accumulation at mid L3 stages (Sup. Fig. 1A-B')."

This description of the model is confusing:

"In our model, as the layers cannot detach from each other, the detachment of cells from

the BM is represented as a thickening of the integrin adhesion layer, where the upper side of this layer in contact with the cells is deformed, while the other side in contact with the BM remains un-deformed and flat. In our model, we observed thickening of the integrin adhesion layer, however, both sides of the layer deformed (Fig. 3C)."

I read these two sentences as contradictory. The first sentence says "In our model... the upper side of this layer in contact with the cells is deformed, which the other side in contact with the BM remains undeformed and flat." The second sentence says "In our model... both sides of the layer deformed." I understand the point that the authors are trying to convey and just ask that they clarify the writing here.

"A recent study has identified a cell-cell adhesion complex containing DE-Cad at the basal-most region of the lateral membrane in different *Drosophila* epithelial tissues, including larval wing imaginal discs, which can be modulated by cell-ECM interactions (Kroeger et al., 2024). This has been proposed to regulate morphogenetic processes, although this needs to be yet demonstrated (Kroeger et al., 2024)."

This information belongs in the introduction. I also think the second sentence has an extra word. I suggest rephrasing both sentences roughly as follows: Adherens junction-like cell-cell adhesion complexes have recently been identified at the basal-most region of the lateral membrane in larval wing imaginal discs and other tissues (Kroeger et al., 2024). While it has been shown that these junctions, which include DE-Cadherin, can be modulated by cell-ECM interactions, their function is yet to be elucidated (Kroeger et al. 2024).

The following section could be expanded:

The loss of basolateral actomyosin accumulation in *wg>betaPS2deltaCyt; betaPS* wing discs prevented us from analyzing experimentally the consequences of reducing integrin levels without affecting basolateral actomyosin accumulation. Therefore, we used our computational model to test this scenario. We found that an increase in basolateral contractility in our computational model, without changing integrin adhesion strength, resulted in some folding (Fig. 6G, Movie S5). However, unlike in the real situation, we found that both sides of the integrin adhesion layer deformed.

This is an exemplary use for the computational model. However, I think the reader needs to better understand why both sides of the integrin adhesion layer deformed and how that result can be interpreted.

2. Significance:

Significance (Required)

This manuscript is an important, timely, and useful addition to the literature. The study it describes addresses a fundamental question in developmental biology: How do epithelial tissues fold? Historically, much attention has been given to epithelial cell dynamics at the (and/or with respect to) the apical tissue surface and there is a large body of literature that describes this work in fly tissues including but not limited to the embryo, pupal notum, and larval wing disc. These authors explore folding at the basal side of the tissue. Their findings should be of interest to a wide group of cell and developmental biologists that study epithelial cell adhesion and epithelial tissue development. I count myself among that group and have particular expertise in flies.

3. How much time do you estimate the authors will need to complete the suggested revisions:

Estimated time to Complete Revisions (Required)

(Decision Recommendation)

Less than 1 month

No

Review #2

1. Evidence, reproducibility and clarity:

Evidence, reproducibility and clarity (Required)

The authors study the actin cytoskeleton and integrins and their impact on epithelial cell shape during *Drosophila* wing development. They find that in a small population of columnar epithelial cells during a specific stage of development when the wing folds, some cells detach from the ECM and contractile actin accumulates basally. Using genetic experiments and computational modelling they present evidence that argues that basal

contractile actin and detachment from the ECM is required for cells to change shape and therefore for tissues to fold.

****Major comments:****

For the most part, the data are presented well and quantified appropriately, and the data support most of the conclusions. I have three comments/questions aimed at improving the paper.

The first relates to data display. Currently, the authors describe the core phenotypes (cell shortening, detachment of integrins from the ECM, integrin downregulation, basal actin and DECaD accumulation) by mostly showing X-Z images of imaginal discs. This means that often only a few cells possess the phenotype in question. It would present a more convincing picture to readers if the authors also presented some magnified X-Y planar images of imaginal discs, i.e. across the basal plane of the tissue, so that readers can better appreciate how many cells are shortening, detaching from the ECM and accumulating basal contractile actin etc. across the tissue.

The second relates to the experiments in Figure 5, i.e. modulation of F-actin accumulation, which is reported to impede basal contractile actin, cell shape changes etc and is concluded to mean that basal contractile actin drives morphogenesis. The authors should either perform further experiments to test the impact of the basal pool of contractile actin specifically on tissue morphogenesis or soften their conclusions on this. Currently, it is impossible to conclude that the changes that are caused by WAVE regulatory complex protein RNAi are specifically due to loss of basal F-actin or total F-actin. Note, I understand that experiments to test this are likely to be very difficult or not possible at all and I also recognise that they have addressed this with computational modelling, and that is why I have suggested that it could be addressed either with experiments or by softening their conclusions.

Finally, related to the point above, how can the authors distinguish between basal contractile actin having an active role in tissue folding/morphogenesis, as opposed to a "supportive role", where it simply ensures epithelial integrity is maintained in response cell detachment from the ECM?

****Referees cross-commenting****

The comments from all reviewers appear to be well aligned and generally appreciate the

value of the paper and how it has been performed and displayed.

The requested experiments and amendments are also reasonable. With respect to major comment 1 from reviewer 3, on the impact of integrin disruption on the apical domain of imaginal discs - this has largely been addressed in a recent paper (PMID: 38134928), which identified basal spot junctions in imaginal discs and also showed that integrin perturbation induced their formation, without having major impacts on the apical domain. It is suggested that the authors refer to this paper when describing that integrin perturbation induces basal e-cad/spot junctions, rather than performing experiments.

2. Significance:

Significance (Required)

I view the paper as both novel and significant, noting the caveat on the difficulty to unequivocally ascribe basal contractile actin to epithelial tissue morphogenesis, as opposed to epithelial tissue support. As such, the authors provide new insights into when and where basal contractile actin appears, that it coincides with loss of integrin/ECM attachment and reveal a potential role for how basal contractile actin might influence tissue morphogenesis. They have performed their experiments carefully and mostly convincingly (note comment above on suggestions to show some data even more clearly). The majority of studies of morphogenesis have focussed on apical contractile actin and overlooked basal contractile actin. Therefore, the paper should be influential and stimulate further studies on basal contractile actin and its role(s) in epithelia in *Drosophila* and other species.

3. How much time do you estimate the authors will need to complete the suggested revisions:

Estimated time to Complete Revisions (Required)

(Decision Recommendation)

Between 1 and 3 months

Yes

Review #3

1. Evidence, reproducibility and clarity:

Evidence, reproducibility and clarity (Required)

Valencia-Expósito and colleagues investigate how cell adhesion impacts the basal folding of tissues (in contrast to the widely-studied formation of apical folds). Using the basal folding of the wing margin in *Drosophila*, the authors find that wing margin cells first become shorter and constrict their base during larval stages, prior to expanding their apex. Also in larval stages, wing margin cells appear to detach from the basement membrane, consistent with a reduction in integrin levels. In parallel, basal F-actin and myosin levels increase. Finite element modelling predicts that the reduction in basal adhesion must precede basolateral contractility, a prediction that the authors validate experimentally. Ectopic integrin downregulation by RNAi induces basal actomyosin and Rho-kinase, accumulation, cell shortening and fold formation. The authors find that in the wing margin and in cells with RNAi-induced downregulation of integrins, E-cadherin forms basal spots, suggesting an antagonistic relationship between the two adhesion systems. In both model and experiments, reducing actin levels without affecting the downregulation of integrins prevented basal folding. Similarly, integrin overexpression prevented the basal accumulation of actin and myosin and the formation of a fold. The authors propose that cell-matrix detachment and basolateral contractility at late larval stages are necessary for basal folding and proper wing formation.

****Major****

1. The authors show that a reduction in integrin levels triggers a basal increase in E-cadherin levels. Does the apical-basal polarity of the cells get reversed or is there a direct effect of integrin on E-cadherin? Staining for apical markers (e.g. Crb) would help to distinguish between these two models.
2. Figures 1G-H, 2 and S1: the authors provide an example of the detachment of a group of cells from the basement membrane, and of changes in integrin levels, F-actin and myosin. However, in these experiments they do not provide a marker for the wing margin cells, so with a single example and without temporal information, it is impossible to confirm that those are the wing margin cells. I am not as worried about Figures 2 and S1, as those are quantified across several disks, but the detachment of cells in Figure 1G-H is not quantified, and only one example is shown. The experiment in Figure 1G-H should be repeated using a marker of wing margin cells, as in Figure 1A-D, to demonstrate that wing

margin cells detach from the basement membrane.

3. Figures 2 and S1: the authors refer to a basolateral accumulation of F-actin and myosin II. However, their arrows point at the basal regions of the cells. They should show exactly where was fluorescence quantified. If the measurements were exclusively on the basal surface of the cells, then they should specifically refer to basal enrichment. This would also have implications for the finite element model, as they implement the "basolateral" actomyosin enrichment as a reduction in the preferred cell height of the basal elements, but it may be that it should be implemented as a reduction in the preferred width of the basal elements if fluorescence was exclusively measured basally.

4. The authors propose that a reduction in integrin levels is sufficient to drive an increase in basal E-cadherin, F-actin and myosin, and to indent tissues basally. The connection between integrins and E-cadherin should be further demonstrated by showing that an E-cadherin knock down in the context of reduced integrin (either in wing margin cells or in cells treated with integrin RNAi) prevents the basal accumulation of F-actin and myosin, and the formation of a basal fold.

****Optional****

1. This is a matter of semantics, but the abstract begins with "During embryogenesis ...", yet none of the presented work is in embryos. I would switch to "During development ...".

2. Figures 5 and S6: Local downregulation of F-actin or myosin (e.g. using optogenetics) would be a cleaner way to show that F-actin contractility drives basal fold formation.

****Typos****

1. (reviewed in (Leptin et al., 1989): missing parenthesis.

2. (Couso et al., 1994), Fig. 1): missing parenthesis.

3. (reviewed in (Brown, 1993; Yee and Hynes, 1993): missing parenthesis.

4. (reviewed in (Streuli, 2009): missing parenthesis.

5. (reviewed in (Clarke and Martin, 2021): missing parenthesis.

6. (reviewed in (Chalut and Paluch, 2016): missing parenthesis.

7. (reviewed in (Mui et al., 2016): missing parenthesis.

8. (reviewed in (Heisenberg and Bellaiche, 2013): missing parenthesis.

2. Significance:

Significance (Required)

This is a well-conducted, solid study, exploring an aspect of tissue folding that is often neglected: matrix adhesion. Furthermore, there are relatively few studies that investigate the mechanisms of basal fold formation, and I anticipate that this work will be widely cited. The results are convincing and well controlled. I just have a couple of questions about how some experiments were conducted, and a couple of additional experiments to solidify the conclusions of the work.

3. How much time do you estimate the authors will need to complete the suggested revisions:

Estimated time to Complete Revisions (Required)

(Decision Recommendation)

Between 1 and 3 months

Yes

Full Revision

Manuscript number: RC-2024-02663

Corresponding author(s): Andrea, Valencia-Expósito

[Please use this template only if the submitted manuscript should be considered by the affiliate journal as a full revision in response to the points raised by the reviewers.]

1. General Statements

Tissue folding is a fundamental process that sculpts simple flat epithelia into complex 3D structures required for proper organ morphogenesis and function. Thus, understanding epithelial folding is crucial to better comprehend epithelia shaping during normal and developmental disease conditions, as well as for the future of tissue engineering and regenerative medicine. During morphogenesis, epithelia can fold either apically or basally. However, while folding of epithelial tissues towards the apical surface has long been studied, the molecular and cellular mechanisms that mediate epithelial folding towards the basal surface are not as well understood.

In this work, using as model system the primordium of the *Drosophila* wing, the wing imaginal disc, we demonstrate opposing roles for cell-cell and cell-extracellular matrix (ECM) adhesion systems during epithelial folding. Thus, while cadherin (E-Cad)-mediated adhesion, linked to actomyosin network, regulates apical folding, a reduction on integrin-dependent adhesion, leading to changes in cell shape, organization of the basal actomyosin cytoskeleton and E-Cad localization, is necessary and sufficient to trigger basal folding. These results suggest that modulation of the cell mechanical landscape through the crosstalk between integrins and cadherins is essential for correct epithelial folding.

The *Drosophila* wing disc has proved a valuable paradigm for much work in the study of the molecular and mechanical mechanisms underlying the acquisition of the 3D shape of epithelia. Besides *Drosophila* wing development, basal tissue folding is required for different morphogenetic events in both vertebrate and invertebrate systems, such as the formation of the midbrain-hindbrain boundary and the optic cup in zebrafish, *Ciona* notochord elongation or Hydra bud formation. In this work, we have discovered a new mechanism underlying precision of basal tissue folding in space and time. These findings will be of interest and significant to a wide group of cell and molecular biologists in the areas of cell adhesion, cytoskeleton and developmental biology, such as the readers of The EMBO Journal. This seems to be well aligned with the comments from all reviewers, who generally appreciate the value and significance of the paper and how it has been performed and displayed, and believe this work should be influential and stimulate further studies on basal contractile actin and its role(s) in epithelia in *Drosophila* and other species.

I would like to take this opportunity to thank the reviewers for the very positive and constructive comments. Most of them aim at improving the paper by adding some clarifications in the text. In addition, a few experiments have been proposed to further sustain our data. These experiments have all been performed and the results further support our conclusions. A full revised version of the article is provided with changes with respect to the original version in red.

This section is mandatory. Please insert a point-by-point reply describing the revisions that were already carried out and included in the transferred manuscript.

Reviewer #1

Evidence, reproducibility and clarity

While infolding from the apical surface (as in the developing *Drosophila* embryo) has been examined previously, infolding from the basal surface is not as well understood and is unlikely to proceed through the same mechanism. How does it work? The authors take a strong approach to attacking this problem: they combine computational modeling with quantitative imaging and use a well-established and appropriate developmental system, the *Drosophila* imaginal wing disc. They show that folding at the basal side of the wing disc is modulated by both cell-matrix adhesion, which decreases locally, and cell-cell adhesion, which increases. The latter adhesion relies on pools of Shotgun (*Drosophila* E-Cadherin) that localize at the basal-most part of the lateral cortical domain. The data and interpretation are straightforward and convincing. I do not have concerns that require additional experimentation.

Minor concerns:

1. Red-green colorblind readers might struggle to interpret some of the confocal micrographs. It should be fairly easy to change the color scheme using FiJi (for example).

We have tried to change the colour of the confocal micrographs but as in most cases there are three channels and, in some cases, up to four, the colour combinations that will suit best colourblind readers are not unfortunately as clear as the actual ones. However, if considered necessary we will be happy to change to change the colour of the images of the main figures.

2. A trivial concern is that the Supplemental Figures weren't numbered. It's relevant to bring this up because in my opinion the model figure (I'm confident this is Sup. Fig. 8) would be a useful addition to the main article (space permitting).

We agree with the reviewer in that it will be useful to have the model figure in the main article. Thus, we have changed it and it is now Fig.9.

3. Please break up the following sentence so that it's easier to read:

"In addition, we found that the distribution of activated non-muscle MyosinII, hereafter MyoII, as detected using an antibody that specifically recognizes the *Drosophila* homolog of the MyoII regulatory light chain, spaghetti squash (sqh), when it is phosphorylated at the activating Ser-21

(pSqh), followed a dynamic similar to that of F-actin, changing from a homogeneous distribution at early stages to a basolateral accumulation at mid L3 stages (Sup. Fig. 1A-B")."

We have broken up the sentence and now it reads as: "Next, we analysed the distribution of activated non-muscle MyosinII, hereafter MyoII. In order to do this, we used an antibody that specifically recognizes the *Drosophila* homolog of the MyoII regulatory light chain, *spaghetti squash* (*sqh*), when it is phosphorylated at the activating Ser-21 (pSqh). We found that pSqh dynamics were similar to that of F-actin, changing from a homogeneous distribution at early stages to a basolateral accumulation at mid L3 stages (Sup. Fig. 1A-B")."

4. This description of the model is confusing:

"In our model, as the layers cannot detach from each other, the detachment of cells from the BM is represented as a thickening of the integrin adhesion layer, where the upper side of this layer in contact with the cells is deformed, while the other side in contact with the BM remains undeformed and flat. In our model, we observed thickening of the integrin adhesion layer, however, both sides of the layer deformed (Fig. 3C)."

I read these two sentences as contradictory. The first sentence says "In our model... the upper side of this layer in contact with the cells is deformed, while the other side in contact with the BM remains undeformed and flat." The second sentence says "In our model... both sides of the layer deformed." I understand the point that the authors are trying to convey and just ask that they clarify the writing here.

The referee is right in that the two sentences seem contradictory and this is because in the second sentence it should say "In our simulation". We have changed the text and now it reads as follows: "In our model, as the layers cannot detach from each other, the detachment of cells from the BM is represented as a thickening of the integrin adhesion layer, where the upper side of this layer in contact with the cells is deformed, while the other side in contact with the BM remains undeformed and flat. However, we found in our simulations that even though a thickening of the integrin adhesion layer was produced, both sides of the layer deformed (Fig. 3C)."

5. "A recent study has identified a cell-cell adhesion complex containing DE-Cad at the basal-most region of the lateral membrane in different *Drosophila* epithelial tissues, including larval wing imaginal discs, which can be modulated by cell-ECM interactions (Kroeger et al., 2024). This has been proposed to regulate morphogenetic processes, although this needs to be yet demonstrated (Kroeger et al., 2024)."

This information belongs in the introduction. I also think the second sentence has an extra word. I suggest rephrasing both sentences roughly as follows: Adherens junction-like cell-cell adhesion complexes have recently been identified at the basal-most region of the lateral membrane in larval wing imaginal discs and other tissues (Kroeger et al., 2024). While it has been shown that these

junctions, which include DE-Cadherin, can be modulated by cell-ECM interactions, their function is yet to be elucidated (Kroeger et al. 2024).

We agree with the reviewer that this information belongs in the introduction and we have moved it there.

6. The following section could be expanded:

The loss of basolateral actomyosin accumulation in *wg>betaPS2deltaCyt; betaPS* wing discs prevented us from analyzing experimentally the consequences of reducing integrin levels without affecting basolateral actomyosin accumulation. Therefore, we used our computational model to test this scenario. We found that an increase in basolateral contractility in our computational model, without changing integrin adhesion strength, resulted in some folding (Fig. 6G, Movie S5). However, unlike in the real situation, we found that both sides of the integrin adhesion layer deformed.

This is an exemplary use for the computational model. However, I think the reader needs to better understand why both sides of the integrin adhesion layer deformed and how that result can be interpreted.

We have added an explanation and a suggestion for this result: “However, unlike in the real situation, we found that both sides of the integrin adhesion layer deformed, as cells cannot detach from the BM. This suggests that even though an increase in basolateral contractility could somehow initiate folding, downregulation of integrin adhesion strength is a prerequisite for proper and complete folding.”

Significance

This manuscript is an important, timely, and useful addition to the literature. The study it describes addresses a fundamental question in developmental biology: How do epithelial tissues fold? Historically, much attention has been given to epithelial cell dynamics at the (and/or with respect to) the apical tissue surface and there is a large body of literature that describes this work in fly tissues including but not limited to the embryo, pupal notum, and larval wing disc. These authors explore folding at the basal side of the tissue. Their findings should be of interest to a wide group of cell and developmental biologists that study epithelial cell adhesion and epithelial tissue development. I count myself among that group and have particular expertise in flies.

Reviewer #2

Evidence, reproducibility and clarity

The authors study the actin cytoskeleton and integrins and their impact on epithelial cell shape during *Drosophila* wing development. They find that in a small population of columnar epithelial

cells during a specific stage of development when the wing folds, some cells detach from the ECM and contractile actin accumulates basally. Using genetic experiments and computational modelling they present evidence that argues that basal contractile actin and detachment from the ECM is required for cells to change shape and therefore for tissues to fold.

Major comments:

For the most part, the data are presented well and quantified appropriately, and the data support most of the conclusions. I have three comments/questions aimed at improving the paper.

1. The first relates to data display. Currently, the authors describe the core phenotypes (cell shortening, detachment of integrins from the ECM, integrin downregulation, basal actin and DECCad accumulation) by mostly showing X-Z images of imaginal discs. This means that often only a few cells possess the phenotype in question. It would present a more convincing picture to readers if the authors also presented some magnified X-Y planar images of imaginal discs, i.e. across the basal plane of the tissue, so that readers can better appreciate how many cells are shortening, detaching from the ECM and accumulating basal contractile actin etc. across the tissue.

To improve this aspect, we have added magnified X-Y planar images of imaginal discs, across the basal plane of the tissue, to Figure 2, which is the first one showing the dynamics of integrin and F-actin expression. We have also added a new Supplementary Figure (Sup. Fig.1), illustrating the different sections, XY, ZX and ZY, shown in the Figures. We believe this will help readers to appreciate the changes under study. If this was not sufficient, we could add X-Y planar images to all main Figures.

2. The second relates to the experiments in Figure 5, i.e. modulation of F-actin accumulation, which is reported to impede basal contractile actin, cell shape changes etc and is concluded to mean that basal contractile actin drives morphogenesis. The authors should either perform further experiments to test the impact of the basal pool of contractile actin specifically on tissue morphogenesis or soften their conclusions on this. Currently, it is impossible to conclude that the changes that are caused by WAVE regulatory complex protein RNAi are specifically due to loss of basal F-actin or total F-actin. Note, I understand that experiments to test this are likely to be very difficult or not possible at all and I also recognise that they have addressed this with computational modelling, and that is why I have suggested that it could be addressed either with experiments or by softening their conclusions.

We agree with the reviewer in that we cannot conclude that the changes that are caused by WAVE regulatory complex protein RNAi are specifically due to loss of basal F-actin or total F-actin. However, as pointed out by the reviewer, at present, this cannot be done experimentally, and this is why we have also addressed it with the computational model. Thus, we have softened the

conclusion of this experiment and now it reads as follows: “This result supports the view that basolateral actomyosin accumulation is most likely necessary to initiate basal folding.”

3. Finally, related to the point above, how can the authors distinguish between basal contractile actin having an active role in tissue folding/morphogenesis, as opposed to a "supportive role", where it simply ensures epithelial integrity is maintained in response to cell detachment from the ECM?

We believe basal contractile actin has an active role in tissue folding/morphogenesis, as opposed to a "supportive role", because when we block formation of basal contractile actin in wing margin cells, by expressing a constitutively active version of the integrins (Fig.7), basal folding is prevented while epithelial integrity does not seem to be notably affected, as judged by the normal morphology of the cell, accumulation of apical actin, which could reflect normal apical-basal polarity, and absence of pycnotic nuclei (a cell death indicator) (Fig.7B). Furthermore, this experimental condition gives rise to a normal looking wing that does not fold properly (Fig.8E-F).

CROSS-REFEREE COMMENTS

The comments from all reviewers appear to be well aligned and generally appreciate the value of the paper and how it has been performed and displayed. The requested experiments and amendments are also reasonable. With respect to major comment 1 from reviewer 3, on the impact of integrin disruption on the apical domain of imaginal discs - this has largely been addressed in a recent paper (PMID: 38134928), which identified basal spot junctions in imaginal discs and also showed that integrin perturbation induced their formation, without having major impacts on the apical domain. It is suggested that the authors refer to this paper when describing that integrin perturbation induces basal e-cad/spot junctions, rather than performing experiments.

We have previously shown that removing integrin function from wing margin cells does not affect their apical-basal polarity, as seen with an antibody against the apical marker aPKC (Martínez-Abarca Millán, A and Martín-Bermudo, M.D. (2023) <https://doi.org/10.3390/cancers15225432>

Significance

I view the paper as both novel and significant, noting the caveat on the difficulty to unequivocally ascribe basal contractile actin to epithelial tissue morphogenesis, as opposed to epithelial tissue support. As such, the authors provide new insights into when and where basal contractile actin appears, that it coincides with loss of integrin/ECM attachment and reveal a potential role for how basal contractile actin might influence tissue morphogenesis. They have performed their experiments carefully and mostly convincingly (note comment above on suggestions to show some data even more clearly). The majority of studies of morphogenesis have focused on apical contractile actin and overlooked basal contractile actin. Therefore, the paper should be influential

and stimulate further studies on basal contractile actin and its role(s) in epithelia in *Drosophila* and other species.

Reviewer #3

Evidence, reproducibility and clarity

Valencia-Expósito and colleagues investigate how cell adhesion impacts the basal folding of tissues (in contrast to the widely-studied formation of apical folds). Using the basal folding of the wing margin in *Drosophila*, the authors find that wing margin cells first become shorter and constrict their base during larval stages, prior to expanding their apex. Also in larval stages, wing margin cells appear to detach from the basement membrane, consistent with a reduction in integrin levels. In parallel, basal F-actin and myosin levels increase. Finite element modelling predicts that the reduction in basal adhesion must precede basolateral contractility, a prediction that the authors validate experimentally. Ectopic integrin downregulation by RNAi induces basal actomyosin and Rho-kinase, accumulation, cell shortening and fold formation. The authors find that in the wing margin and in cells with RNAi-induced downregulation of integrins, E-cadherin forms basal spots, suggesting an antagonistic relationship between the two adhesion systems. In both model and experiments, reducing actin levels without affecting the downregulation of integrins prevented basal folding. Similarly, integrin overexpression prevented the basal accumulation of actin and myosin and the formation of a fold. The authors propose that cell-matrix detachment and basolateral contractility at late larval stages are necessary for basal folding and proper wing formation.

MAJOR

1. The authors show that a reduction in integrin levels triggers a basal increase in E-cadherin levels. Does the apical-basal polarity of the cells get reversed or is there a direct effect of integrin on E-cadherin? Staining for apical markers (e.g. Crb) would help to distinguish between these two models.

We have previously shown that removing integrin function from wing margin cells does not affect their apical-basal polarity, as seen with an antibody against the apical marker aPKC (Martínez-Abarca Millán, A and Martín-Bermudo, M.D. (2023) <https://doi.org/10.3390/cancers15225432>. We have added this information in the Results section.

2. Figures 1G-H, 2 and S1: the authors provide an example of the detachment of a group of cells from the basement membrane, and of changes in integrin levels, F-actin and myosin. However, in these experiments they do not provide a marker for the wing margin cells, so with a single example and without temporal information, it is impossible to confirm that those are the wing margin cells. I am not as worried about Figures 2 and S1, as those are quantified across several disks, but the

detachment of cells in Figure 1G-H is not quantified, and only one example is shown. The experiment in Figure 1G-H should be repeated using a marker of wing margin cells, as in Figure 1A-D, to demonstrate that wing margin cells detach from the basement membrane.

We have incorporated in Fig.1 an image showing a wing disc stained with antibodies against Wingless, a marker for wing margin cells, and Perlecan, to label the BM. This image demonstrates that wing margin cells detach from the BM.

3. Figures 2 and S1: the authors refer to a basolateral accumulation of F-actin and myosin II. However, their arrows point at the basal regions of the cells. They should show exactly where was fluorescence quantified. If the measurements were exclusively on the basal surface of the cells, then they should specifically refer to basal enrichment. This would also have implications for the finite element model, as they implement the "basolateral" actomyosin enrichment as a reduction in the preferred cell height of the basal elements, but it may be that it should be implemented as a reduction in the preferred width of the basal elements if fluorescence was exclusively measured basally.

The arrows point to the cells of the wing epithelium where we appreciate changes in the levels of the different markers. Fluorescence of the different markers has been measured along the basolateral side of the cells and the yellow and orange rectangles determine the region measured, where the height of the rectangle represents the portion of the lateral side of the cells that has been measured, up to 7 mm from the most basal side of the cells. This has now been indicated in the Methods Details section.

4. The authors propose that a reduction in integrin levels is sufficient to drive an increase in basal E-cadherin, F-actin and myosin, and to indent tissues basally. The connection between integrins and E-cadherin should be further demonstrated by showing that an E-cadherin knock down in the context of reduced integrin (either in wing margin cells or in cells treated with integrin RNAi) prevents the basal accumulation of F-actin and myosin, and the formation of a basal fold.

We have reduced E-cadherin expression specifically in wing margin cells and found, as the reviewer expected, that this prevented F-actin basal accumulation and the formation of a basal fold. We have incorporated this result in the results section and in a new Fig.4.

OPTIONAL

1. This is a matter of semantics, but the abstract begins with "During embryogenesis ...", yet none of the presented work is in embryos. I would switch to "During development ...".

Thanks, we have now changed embryogenesis for development.

2. Figures 5 and S6: Local downregulation of F-actin or myosin (e.g. using optogenetics) would be a cleaner way to show that F-actin contractility drives basal fold formation.

Optogenetics have been used in the *Drosophila* embryos to manipulate myosin pattern and analyse cell behaviour during embryonic axis elongation, by either increasing (optoGEF) or decreasing (optoGAP) Rho 1 activity (Herrera-Perez et al. Biophysical J. 2021). In this context, the changes in myosin and cell shape occur within a period of 5-10 mins and can therefore be measured while the effects of the optogenetic perturbations last. However, the changes in actomyosin levels and cell shape that drives basal fold formation occur over periods of hours, hampering the possibility to use these optogenetic tools to analyse this process. In the future, it will be interesting to explore other possibilities.

TYPOS

1. (reviewed in (Leptin et al., 1989): missing parenthesis.
2. (Couso et al., 1994), Fig. 1): missing parenthesis.
3. (reviewed in (Brown, 1993; Yee and Hynes, 1993): missing parenthesis.
4. (reviewed in (Streuli, 2009): missing parenthesis.
5. (reviewed in (Clarke and Martin, 2021): missing parenthesis.
6. (reviewed in (Chalut and Paluch, 2016): missing parenthesis.
- 7.(reviewed in (Mui et al., 2016): missing parenthesis.
8. (reviewed in (Heisenberg and Bellaiche, 2013): missing parenthesis.

Many thanks for pointing out these typos, most related to the formatting process, they have now all been corrected.

Reviewer #3

Significance

This is a well-conducted, solid study, exploring an aspect of tissue folding that is often neglected: matrix adhesion. Furthermore, there are relatively few studies that investigate the mechanisms of basal fold formation, and I anticipate that this work will be widely cited. The results are convincing and well controlled. I just have a couple of questions about how some experiments were conducted, and a couple of additional experiments to solidify the conclusions of the work.

Dear Dr. Martin-Bermudo,

Thank you for submitting your revised Review Commons manuscript to The EMBO Journal. I have now gone through your manuscript, the reviewers' comments and your response to them. Based on the positive assessments by the reviewers and your response to their main comments, which I judge to be very reasonable, I will be happy to accept the manuscript for publication in The EMBO Journal after its reformatting along the guidelines included below and in the attached document.

Upon resubmission, please provide an institutional email address to the co-corresponding author Andrea Valencia-Expósito.

Furthermore, regarding the point raised by reviewer #1 on the use of colour blindness-friendly colour scheme, please consider consulting these guidelines (scroll down for the section on microscopy images): <https://www.nki.nl/about-us/responsible-research/guidelines-color-blind-friendly-figures/>

Please feel free to contact me if you have any further questions regarding this final editorial revision. You can use the link below to upload the revised files.

Thank you for the opportunity to consider your work for publication, and I look forward to receiving the revised manuscript.

With best regards,

Ieva

Revision to The EMBO Journal should be submitted online within 90 days, unless an extension has been requested and approved by the editor; please click on the link below to submit the revision online before 19th Feb 2025:

Link Not Available

Rev_Com_number: RC-2024-02663

New_manu_number: EMBOJ-2024-119652-T

Corr_author: Martin-Bermudo

Title: Local weakening of cell-ECM adhesion triggers basal epithelia tissue folding

Point-by-point Response to Reviewers

Reviewer #1

Evidence, reproducibility and clarity

While infolding from the apical surface (as in the developing *Drosophila* embryo) has been examined previously, infolding from the basal surface is not as well understood and is unlikely to proceed through the same mechanism. How does it work? The authors take a strong approach to attacking this problem: they combine computational modeling with quantitative imaging and use a well-established and appropriate developmental system, the *Drosophila* imaginal wing disc. They show that folding at the basal side of the wing disc is modulated by both cell-matrix adhesion, which decreases locally, and cell-cell adhesion, which increases. The latter adhesion relies on pools of Shotgun (*Drosophila* E-Cadherin) that localize at the basal-most part of the lateral cortical domain. The data and interpretation are straightforward and convincing. I do not have concerns that require additional experimentation.

Minor concerns:

1. Red-green colorblind readers might struggle to interpret some of the confocal micrographs. It should be fairly easy to change the color scheme using FiJi (for example).

We have changed the colour of the confocal micrographs to best suit colourblind readers.

2. A trivial concern is that the Supplemental Figures weren't numbered. It's relevant to bring this up because in my opinion the model figure (I'm confident this is Sup. Fig. 8) would be a useful addition to the main article (space permitting).

We agree with the reviewer in that it will be useful to have the model figure in the main article. Thus, we have changed it and it is now Fig.9.

3. Please break up the following sentence so that it's easier to read:

"In addition, we found that the distribution of activated non-muscle MyosinII, hereafter MyoII, as detected using an antibody that specifically recognizes the *Drosophila* homolog of the MyoII regulatory light chain, spaghetti squash (sqh), when it is phosphorylated at the activating Ser-21

(pSqh), followed a dynamic similar to that of F-actin, changing from a homogeneous distribution at early stages to a basolateral accumulation at mid L3 stages (Sup. Fig. 1A-B").

We have broken up the sentence and now it reads as: "Next, we analysed the distribution of activated non-muscle MyosinII, hereafter MyoII. In order to do this, we used an antibody that specifically recognizes the *Drosophila* homolog of the MyoII regulatory light chain, *spaghetti squash* (*sqh*), when it is phosphorylated at the activating Ser-21 (pSqh). We found that pSqh dynamics were similar to that of F-actin, changing from a homogeneous distribution at early stages to a basolateral accumulation at mid L3 stages.

4. This description of the model is confusing:

"In our model, as the layers cannot detach from each other, the detachment of cells from the BM is represented as a thickening of the integrin adhesion layer, where the upper side of this layer in contact with the cells is deformed, while the other side in contact with the BM remains undeformed and flat. In our model, we observed thickening of the integrin adhesion layer, however, both sides of the layer deformed (Fig. 3C)."

I read these two sentences as contradictory. The first sentence says "In our model... the upper side of this layer in contact with the cells is deformed, while the other side in contact with the BM remains undeformed and flat." The second sentence says "In our model... both sides of the layer deformed." I understand the point that the authors are trying to convey and just ask that they clarify the writing here.

The referee is right in that the two sentences seem contradictory and this is because in the second sentence it should say "In our simulation". We have changed the text and now it reads as follows: "In our model, as the layers cannot detach from each other, the detachment of cells from the BM is represented as a thickening of the integrin adhesion layer, where the upper side of this layer in contact with the cells is deformed, while the other side in contact with the BM remains undeformed and flat. However, we found in our simulations that even though a thickening of the integrin adhesion layer was produced, both sides of the layer deformed."

5. "A recent study has identified a cell-cell adhesion complex containing DE-Cad at the basal most-region of the lateral membrane in different *Drosophila* epithelial tissues, including larval wing imaginal discs, which can be modulated by cell-ECM interactions (Kroeger et al., 2024). This has been proposed to regulate morphogenetic processes, although this needs to be yet demonstrated (Kroeger et al., 2024)."

This information belongs in the introduction. I also think the second sentence has an extra word. I suggest rephrasing both sentences roughly as follows: Adherens junction-like cell-cell adhesion complexes have recently been identified at the basal-most region of the lateral membrane in larval wing imaginal discs and other tissues (Kroeger et al., 2024). While it has been shown that these junctions, which include DE-Cadherin, can be modulated by cell-ECM interactions, their function is yet to be elucidated (Kroeger et al. 2024).

We agree with the reviewer that this information belongs in the introduction and we have moved it there.

6. The following section could be expanded:

The loss of basolateral actomyosin accumulation in *wg>betaPS2deltaCyt; betaPS* wing discs prevented us from analyzing experimentally the consequences of reducing integrin levels without affecting basolateral actomyosin accumulation. Therefore, we used our computational model to test this scenario. We found that an increase in basolateral contractility in our computational model, without changing integrin adhesion strength, resulted in some folding (Fig. 6G, Movie S5). However, unlike in the real situation, we found that both sides of the integrin adhesion layer deformed.

This is an exemplary use for the computational model. However, I think the reader needs to better understand why both sides of the integrin adhesion layer deformed and how that result can be interpreted.

We have added an explanation and a suggestion for this result: "However, unlike in the real situation, we found that both sides of the integrin adhesion layer deformed, as cells cannot detach from the BM. This suggests that even though an increase in basolateral contractility could

somehow initiate folding, downregulation of integrin adhesion strength is a prerequisite for proper and complete folding.”

Significance

This manuscript is an important, timely, and useful addition to the literature. The study it describes addresses a fundamental question in developmental biology: How do epithelial tissues fold? Historically, much attention has been given to epithelial cell dynamics at the (and/or with respect to) the apical tissue surface and there is a large body of literature that describes this work in fly tissues including but not limited to the embryo, pupal notum, and larval wing disc. These authors explore folding at the basal side of the tissue. Their findings should be of interest to a wide group of cell and developmental biologists that study epithelial cell adhesion and epithelial tissue development. I count myself among that group and have particular expertise in flies.

Reviewer #2

Evidence, reproducibility and clarity

The authors study the actin cytoskeleton and integrins and their impact on epithelial cell shape during *Drosophila* wing development. They find that in a small population of columnar epithelial cells during a specific stage of development when the wing folds, some cells detach from the ECM and contractile actin accumulates basally. Using genetic experiments and computational modelling they present evidence that argues that basal contractile actin and detachment from the ECM is required for cells to change shape and therefore for tissues to fold.

Major comments:

For the most part, the data are presented well and quantified appropriately, and the data support most of the conclusions. I have three comments/questions aimed at improving the paper.

1. The first relates to data display. Currently, the authors describe the core phenotypes (cell shortening, detachment of integrins from the ECM, integrin downregulation, basal actin and DECaad accumulation) by mostly showing X-Z images of imaginal discs. This means that often only a few cells possess the phenotype in question. It would present a more convincing picture to readers if the authors also presented some magnified X-Y planar images of imaginal discs, i.e. across the basal plane of the tissue, so that readers can better appreciate how many cells are shortening, detaching from the ECM and accumulating basal contractile actin etc. across the tissue.

To improve this aspect, we have added magnified X-Y planar images of imaginal discs, across the basal plane of the tissue, to Figure 2, which is the first one showing the dynamics of integrin and F-actin expression. We have also added a new Supplementary Figure (Supplementary Fig.1), illustrating the different sections, XY, ZX and ZY, shown in the Figures. We believe this will help readers to appreciate the changes under study.

2. The second relates to the experiments in Figure 5, i.e. modulation of F-actin accumulation, which is reported to impede basal contractile actin, cell shape changes etc and is concluded to mean that basal contractile actin drives morphogenesis. The authors should either perform further experiments to test the impact of the basal pool of contractile actin specifically on tissue

morphogenesis or soften their conclusions on this. Currently, it is impossible to conclude that the changes that are caused by WAVE regulatory complex protein RNAi are specifically due to loss of basal F-actin or total F-actin. Note, I understand that experiments to test this are likely to be very difficult or not possible at all and I also recognise that they have addressed this with computational modelling, and that is why I have suggested that it could be addressed either with experiments or by softening their conclusions.

We agree with the reviewer in that we cannot conclude that the changes that are caused by WAVE regulatory complex protein RNAi are specifically due to loss of basal F-actin or total F-actin. However, as pointed out by the reviewer, at present, this cannot be done experimentally, and this is why we have also addressed it with the computational model. Thus, we have softened the conclusion of this experiment and now it reads as follows: “This result supports the view that basolateral actomyosin accumulation is most likely necessary to initiate basal folding.”

3. Finally, related to the point above, how can the authors distinguish between basal contractile actin having an active role in tissue folding/morphogenesis, as opposed to a "supportive role", where it simply ensures epithelial integrity is maintained in response to cell detachment from the ECM?

We believe basal contractile actin has an active role in tissue folding/morphogenesis, as opposed to a "supportive role", because when we block formation of basal contractile actin in wing margin cells, by expressing a constitutively active version of the integrins, basal folding is prevented while epithelial integrity does not seem to be notably affected, as judged by the normal morphology of the cell, accumulation of apical actin, which could reflect normal apical-basal polarity, and absence of pycnotic nuclei (a cell death indicator). Furthermore, this experimental condition gives rise to a normal looking wing that does not fold properly.

CROSS-REFEREE COMMENTS

The comments from all reviewers appear to be well aligned and generally appreciate the value of the paper and how it has been performed and displayed. The requested experiments and amendments are also reasonable. With respect to major comment 1 from reviewer 3, on the impact

of integrin disruption on the apical domain of imaginal discs - this has largely been addressed in a recent paper (PMID: 38134928), which identified basal spot junctions in imaginal discs and also showed that integrin perturbation induced their formation, without having major impacts on the apical domain. It is suggested that the authors refer to this paper when describing that integrin perturbation induces basal e-cad/spot junctions, rather than performing experiments.

Significance

I view the paper as both novel and significant, noting the caveat on the difficulty to unequivocally ascribe basal contractile actin to epithelial tissue morphogenesis, as opposed to epithelial tissue support. As such, the authors provide new insights into when and where basal contractile actin appears, that it coincides with loss of integrin/ECM attachment and reveal a potential role for how basal contractile actin might influence tissue morphogenesis. They have performed their experiments carefully and mostly convincingly (note comment above on suggestions to show some data even more clearly). The majority of studies of morphogenesis have focused on apical contractile actin and overlooked basal contractile actin. Therefore, the paper should be influential and stimulate further studies on basal contractile actin and its role(s) in epithelia in *Drosophila* and other species.

Reviewer #3

Evidence, reproducibility and clarity

Valencia-Expósito and colleagues investigate how cell adhesion impacts the basal folding of tissues (in contrast to the widely-studied formation of apical folds). Using the basal folding of the wing margin in *Drosophila*, the authors find that wing margin cells first become shorter and constrict their base during larval stages, prior to expanding their apex. Also in larval stages, wing margin cells appear to detach from the basement membrane, consistent with a reduction in integrin levels. In parallel, basal F-actin and myosin levels increase. Finite element modelling predicts that the reduction in basal adhesion must precede basolateral contractility, a prediction that the authors validate experimentally. Ectopic integrin downregulation by RNAi induces basal actomyosin and Rho-kinase, accumulation, cell shortening and fold formation. The authors find that in the wing margin and in cells with RNAi-induced downregulation of integrins, E-cadherin forms basal spots, suggesting an antagonistic relationship between the two adhesion systems. In both model and experiments, reducing actin levels without affecting the downregulation of integrins prevented basal folding. Similarly, integrin overexpression prevented the basal accumulation of actin and myosin and the formation of a fold. The authors propose that cell-matrix detachment and basolateral contractility at late larval stages are necessary for basal folding and proper wing formation.

MAJOR

1. The authors show that a reduction in integrin levels triggers a basal increase in E-cadherin levels. Does the apical-basal polarity of the cells get reversed or is there a direct effect of integrin on E-cadherin? Staining for apical markers (e.g. Crb) would help to distinguish between these two models.

We have previously shown that removing integrin function from wing margin cells does not affect their apical-basal polarity, as seen with an antibody against the apical marker aPKC (Martínez-Abarca Millán, A and Martín-Bermudo, M.D. (2023) <https://doi.org/10.3390/cancers15225432>).

2. Figures 1G-H, 2 and S1: the authors provide an example of the detachment of a group of cells from the basement membrane, and of changes in integrin levels, F-actin and myosin. However, in these experiments they do not provide a marker for the wing margin cells, so with a single example and without temporal information, it is impossible to confirm that those are the wing margin cells. I am not as worried about Figures 2 and S1, as those are quantified across several disks, but the detachment of cells in Figure 1G-H is not quantified, and only one example is shown. The experiment in Figure 1G-H should be repeated using a marker of wing margin cells, as in Figure 1A-D, to demonstrate that wing margin cells detach from the basement membrane.

We have incorporated in Fig.1 an image showing a wing disc stained with antibodies against Wingless, a marker for wing margin cells, and Perlecan, to label the BM. This image demonstrates that wing margin cells detach from the BM.

3. Figures 2 and S1: the authors refer to a basolateral accumulation of F-actin and myosin II. However, their arrows point at the basal regions of the cells. They should show exactly where was fluorescence quantified. If the measurements were exclusively on the basal surface of the cells, then they should specifically refer to basal enrichment. This would also have implications for the finite element model, as they implement the "basolateral" actomyosin enrichment as a reduction in the preferred cell height of the basal elements, but it may be that it should be implemented as a reduction in the preferred width of the basal elements if fluorescence was exclusively measured basally.

The arrows point to the cells of the wing epithelium where we appreciate changes in the levels of the different markers. Fluorescence of the different markers has been measured along the basolateral side of the cells and the yellow and orange rectangles determine the region measured, where the height of the rectangle represents the portion of the lateral side of the cells that has been measured, up to 7 μm from the most basal side of the cells. This has now been indicated in the Methods Details section.

4. The authors propose that a reduction in integrin levels is sufficient to drive an increase in basal E-cadherin, F-actin and myosin, and to indent tissues basally. The connection between integrins

and E-cadherin should be further demonstrated by showing that an E-cadherin knock down in the context of reduced integrin (either in wing margin cells or in cells treated with integrin RNAi) prevents the basal accumulation of F-actin and myosin, and the formation of a basal fold.

We have reduced E-cadherin expression specifically in wing margin cells and found, as the reviewer expected, that this prevented F-actin basal accumulation and the formation of a basal fold. We have incorporated this result in the results section and in a new Fig.4.

OPTIONAL

1. This is a matter of semantics, but the abstract begins with "During embryogenesis ...", yet none of the presented work is in embryos. I would switch to "During development ...".

Thanks, we have now changed embryogenesis for development.

2. Figures 5 and S6: Local downregulation of F-actin or myosin (e.g. using optogenetics) would be a cleaner way to show that F-actin contractility drives basal fold formation.

Optogenetics have been used in the *Drosophila* embryos to manipulate myosin pattern and analyse cell behaviour during embryonic axis elongation, by either increasing (optoGEF) or decreasing (optoGAP) Rho 1 activity (Herrera-Perez et al. Biophysical J. 2021). In this context, the changes in myosin and cell shape occur within a period of 5-10 mins and can therefore be measured while the effects of the optogenetic perturbations last. However, the changes in actomyosin levels and cell shape that drives basal fold formation occur over periods of hours, hampering the possibility to use these optogenetic tools to analyse this process. In the future, it will be interesting to explore other possibilities.

TYPOS

1. (reviewed in (Leptin et al., 1989): missing parenthesis.
2. (Couso et al., 1994), Fig. 1): missing parenthesis.
3. (reviewed in (Brown, 1993; Yee and Hynes, 1993): missing parenthesis.

4. (reviewed in (Streuli, 2009): missing parenthesis.
5. (reviewed in (Clarke and Martin, 2021): missing parenthesis.
6. (reviewed in (Chalut and Paluch, 2016): missing parenthesis.
- 7.(reviewed in (Mui et al., 2016): missing parenthesis.
8. (reviewed in (Heisenberg and Bellaiche, 2013): missing parenthesis.

Many thanks for pointing out these typos, most related to the formatting process, they have now all been corrected.

Reviewer #3

Significance

This is a well-conducted, solid study, exploring an aspect of tissue folding that is often neglected: matrix adhesion. Furthermore, there are relatively few studies that investigate the mechanisms of basal fold formation, and I anticipate that this work will be widely cited. The results are convincing and well controlled. I just have a couple of questions about how some experiments were conducted, and a couple of additional experiments to solidify the conclusions of the work.

Dear Lola,

Thank you for submitting a reformatted version of your manuscript. I sincerely apologise for the delay in the processing of your revised manuscript due to the holiday period, followed by my absence from the office due to an illness in the family.

I have now gone through the revised version, and I am afraid that there remain a few formatting aspects as outlined below that still need to be implemented in the manuscript before its acceptance:

1. Please make sure that the order of the sections in the manuscript is as follows: abstract, introduction, results, discussion, materials & methods, data availability section, acknowledgments, disclosure statement and competing interests, references, main figure legends, tables, expanded figure legends.
2. We are missing the ORCID iD for the co-corresponding author Andrea Valencia-Expósito. In order to link the ORCID iD to the account in our manuscript tracking system, the author in question has to do the following:
 - Click the 'Modify Profile' link at the bottom of your homepage in our system.
 - On the next page you will see a box halfway down the page titled ORCID*. Below this box is red text reading 'To Register/Link to ORCID, click here'. Please follow that link: you will be taken to ORCID where you can log in to your account (or create an account if you don't have one)
 - You will then be asked to authorise Wiley to access your ORCID information. Once you have approved the linking, you will be brought back to our manuscript system.Unfortunately, we cannot do this linking on the author's behalf for security reasons.
1. CRediT has replaced the traditional author contributions section because it offers a systematic, machine-readable author contributions format that allows for more effective research assessment. Please remove the Authors Contributions from the manuscript and use the free text boxes beneath each contributing author's name in our online submission system to add specific details on the author's contribution. More information is available in our guide to authors.
2. Please rename "Declaration of interests" section into "Disclosure and competing interests statement" (further info: <https://www.embopress.org/page/journal/14602075/authorguide#conflictsofinterest>).
3. Please move "Data Availability" section to the end of Methods. Since no data deposition in external databases is needed for this paper, please state in this section: "This study includes no data deposited in external repositories". Please also remove the sections on resource and materials availability. More information about the format of this section can be found here: <https://www.embopress.org/page/journal/14602075/authorguide#dataavailability>
4. Figure panel 5I is not mentioned in the manuscript text, please add the corresponding callout.
5. Please rename the movies into Movie EV1-EV5 and update the callouts accordingly. Please upload movies as pearacte files per each movie. The legends should be removed from the manuscript text file and zipped with each movie file. Further information is available here: <https://www.embopress.org/page/journal/14602075/authorguide#expandedview>
6. Please rename the file with Supplementary figures into "Appendix" and add a table of contents with page numbers to the first page. The figures should be renamed "Appendix Figure S1 - S4" and the callouts updated accordingly. The Appendix figure legends should be removed from the manuscript and added to the Appendix, ideally underneath the corresponding figure.
7. Please rename "Star Methods" section into "Methods".
8. Please remove the Key resources table from the manuscript text file.
9. Please zip and upload numerical and image source data files for the main figures as one file per figure.
10. In our standard source data check, we have noted unexplained numerical duplications in the source data for figures 7D and EV2D. I have attached the corresponding files with the detected duplications labelled in colour. Please take a look and correct if needed. A brief explanation would be very helpful.
11. Our data editors have flagged the following issues in figure legends that need correcting:
 - Please provide the exact p values in the legends of figures 1E, F; 2D, E, F; 3H; 5B, D, F, H, I; 6C, E, G, H; 7C, D, E; EV1 E; EV2 D; EV3 D-F; EV4 B, D, F, H;.
 - 2. Please note that in figures 1E, F; 3H, 5B, D, F, H; EV1 E; EV2 D, EV3 D-F; EV4 B, F, H there is a mismatch between the annotated p values in the figure legend and the annotated p values in the figure file that should be corrected.
 - 3. Please indicate what */ **/ ***/ **** represents in the legends of figures 4C-E; if this represents p values, please indicate the statistical test used and specify the exact p value.
 - 4. Please describe the number and nature of replicates in the legends of figures 4C-E.
 - 5. Please define the error bars in the legends of figures 1E, F; 2D-F; 3G, H; 4C-E; 5B, D, F, H, I; 6C, E, G, H; 7C, D, E; EV1 E; EV2 D, EV3 D-F; EV4 B, F, H.
12. Papers published in The EMBO Journal are accompanied online by a 'Synopsis' to enhance discoverability of the manuscript. It consists of A) a short (1-2 sentences) summary of the findings and their significance, B) 3-4 bullet points highlighting key results and C) a synopsis image that is 550x300-600 pixels large (width x height, jpeg or png format). You can either show a model or key data in the synopsis image. Please note that the image size is rather small and that text needs to be readable at the final size. Please send us this information together with the revised manuscript.

Please feel free to contact me if have any questions regarding this final revision. Thank you again for giving us the chance to consider your manuscript for The EMBO Journal. I look forward to receiving the final version!

With best regards,

Ieva

Revision to The EMBO Journal should be submitted online within 90 days, unless an extension has been requested and approved by the editor; please click on the link below to submit the revision online before 23rd Apr 2025:

Link Not Available

Rev_Com_number: RC-2024-02663

New_manu_number: EMBOJ-2024-119652R

Corr_author: Martin-Bermudo

Title: Local weakening of cell-ECM adhesion triggers basal epithelia tissue folding

The authors addressed the remaining editorial issues.

Dear Lola,

Thank you for submitting the final revised version and addressing the remaining editorial points. I am now pleased to inform you that your manuscript has been accepted for publication. Congratulations on a great study!

Before we forward your manuscript to our publishers, I would like to propose some edits in the manuscript title, abstract and synopsis (please also see the attached file). I have also chosen a short blurb that will accompany the title of your manuscript in our online system. Please take a look and let me know if any corrections are needed.

Title:

Local weakening of cell-extracellular matrix adhesion triggers basal epithelial tissue folding

Blurb:

In the *Drosophila* wing disc epithelium, basal folding is induced by localized downregulation of integrin, which triggers reorganization of E-cadherin and actomyosin networks.

Synopsis

Epithelial folding is a fundamental process that sculpts flat epithelia into 3D structures. This study that integrin-dependent adhesion is lost at specific sites, inducing cell shape changes and initiating basal epithelial folding in the *Drosophila* wing imaginal disc epithelium.

- A local weakening of integrin-dependent cell-extracellular matrix (ECM) adhesion is necessary and sufficient to trigger folding of the epithelium towards the basal side.
- The localized downregulation of integrin levels triggers an increase in basolateral contractility and a basal enrichment of E-cadherin.
- The resulting reorganization of E-cadherin and actomyosin networks lead to the cell shape changes which are required for epithelial folding.

If you have any questions, please do not hesitate to contact the Editorial Office. Thank you for this contribution to The EMBO Journal and congratulations on a nice study!

With best wishes,

leva

leva Gailite, PhD
Senior Scientific Editor
The EMBO Journal
Meyerohofstrasse 1
D-69117 Heidelberg
Tel: +4962218891309
i.gailite@embojournal.org

>>> Please note that it is The EMBO Journal policy for the transcript of the editorial process (containing referee reports and your response letter) to be published as an online supplement to each paper. If you do NOT want this, you will need to inform the

Editorial Office via email immediately. More information is available here: https://www.embopress.org/transparent-process#Review_Process

Rev_Com_number: RC-2024-02663

New_manu_number: EMBOJ-2024-119652R1

Corr_author: Martin-Bermudo

Title: Local weakening of cell-ECM adhesion triggers basal epithelia tissue folding